# Reconciling Modern Deep Learning with Traditional Optimization Analyses: The Intrinsic Learning Rate

**Zhiyuan Li**[*]
Princeton University
zhiyuanli@cs.princeton.edu

**Kaifeng Lyu** [*]
Tsinghua University
vfleaking@gmail.com

**Sanjeev Arora**
Princeton University & IAS
arora@cs.princeton.edu

## Abstract

Recent works (e.g., (Li and Arora, 2020)) suggest that the use of popular normalization schemes (including Batch Normalization) in today's deep learning can move it far from a traditional optimization viewpoint, e.g., use of exponentially increasing learning rates. The current paper highlights other ways in which behavior of normalized nets departs from traditional viewpoints, and then initiates a formal framework for studying their mathematics via suitable adaptation of the conventional framework namely, modeling SGD-induced training trajectory via a suitable stochastic differential equation (SDE) with a noise term that captures gradient noise. This yields: (a) A new "intrinsic learning rate" parameter that is the product of the normal learning rate $\eta$ and weight decay factor $\lambda$. Analysis of the SDE shows how the effective speed of learning varies and equilibrates over time under the control of intrinsic LR. (b) A challenge—via theory and experiments—to popular belief that good generalization requires large learning rates at the start of training. (c) New experiments, backed by mathematical intuition, suggesting the number of steps to equilibrium (in function space) scales as the inverse of the intrinsic learning rate, as opposed to the exponential time convergence bound implied by SDE analysis. We name it the *Fast Equilibrium Conjecture* and suggest it holds the key to why Batch Normalization is effective.

## 1   Introduction

The training of modern neural networks involves Stochastic Gradient Descent (SGD) with an appropriate learning rate schedule. The formula of SGD with weight decay can be written as:
$$\boldsymbol{w}_{t+1} \leftarrow (1 - \eta_t \lambda) \boldsymbol{w}_t - \eta_t \nabla \mathcal{L}_t(\boldsymbol{w}_t),$$
where $\lambda$ is the weight decay factor (or $L^2$-regularization coefficient), $\eta_t$ and $\nabla \mathcal{L}_t(\boldsymbol{w}_t)$ are the learning rate and batch gradient at the $t$-th iteration.

Traditional analysis shows that SGD approaches a stationary point of the training loss if the learning rate is set to be sufficiently small depending on the smoothness constant and noise scale. In this viewpoint, if we reduce the learning rate by a factor 10, the end result is the same, and just takes 10 times as many steps. SGD with very tiny step sizes can be thought of as Gradient Descent (GD) (i.e., gradient descent with full gradient), which in the limit of infinitesimal step size approaches Gradient Flow (GF).

However, it is well-known that using only small learning rates or large batch sizes (while fixing other hype-parameters) may lead to worse generalization (Bengio, 2012; Keskar et al., 2017). From this one concludes that finite (not too small) learning rate —alternatively, noise in the gradient estimate, or small batch sizes— play an important role in generalization, and many authors have suggested that the noise helps avoid sharp minima (Hochreiter and Schmidhuber, 1997; Keskar et al., 2017; Li et al.,

---

[*]These authors contribute equally.

2018; Izmailov et al., 2018; He et al., 2019). Formal understanding of the effect involves modeling SGD via a Stochastic Differential Equation (SDE) in the continuous time limit (Li and Tai, 2019):

$$dW_t = -\eta(t)\lambda W_t dt - \eta(t)\nabla\mathcal{L}(W_t)dt + \eta(t)\Sigma_{W_t}^{1/2}dB_t,$$

where $\Sigma_w$ is the covariance matrix of the noise at $W_t = w$. Several works have adopted this SDE view and given some rigorous analysis of the effect of noise (Smith and Le, 2018; Chaudhari and Soatto, 2018; Shi et al., 2020).

While this SDE view is well-established, we will note in this paper that the past works (both theory and experiments) often draw intuition from shallow nets and do not help understand modern architectures, which can be very deep and crucially rely upon normalization schemes such as Batch Normalization (BN) (Ioffe and Szegedy, 2015), Group Normalization (GN) (Wu and He, 2018), Weight Normalization (WN) (Salimans and Kingma, 2016). We will discuss in Section 4 that these normalization schemes are incompatible to the traditional view points in the following senses. First, normalization makes the loss provably non-smooth around origin, so GD could behave (and does behave, in our experiments) significantly differently from its continuous counterpart, GF, if weight decay is turned on. For example, GD may oscillate between zero loss and high loss and thus cannot persistently achieve perfect interpolation. Second, there is experimental evidence suggesting that the above SDE may be far from mixing for normalized networks in normal training budgets. Lastly, assumptions about the noise in the gradient being a fixed Gaussian turn out to be unrealistic.

In this work, we incorporate effects of normalization in the SDE view to study the complex interaction between BN, weight decay, and learning rate schedule. We particularly focus on Step Decay, one of the most commonly-used learning rate schedules. Here the training process is divided into several phases $1, \ldots, K$. In each phase $i$, the learning rate $\eta_t$ is kept as a constant $\bar{\eta}_i$, and the constants $\bar{\eta}_i$ are decreasing with the phase number $i$. Our main experimental observation is the following one, which is formally stated as a conjecture in the context of SDE in Section 5.2.

**Observation 1.1.** If trained for sufficiently long time during some phase $i$ $(1 \le i \le K)$, a neural network with BN and weight decay will eventually reach an equilibrium distribution in the function space. This equilibrium only depends on the product $\bar{\eta}_i\lambda$, and is independent of the history in the previous phases. Furthermore, the time that the neural net stays at this equilibrium will not affect its future performance.

**Our contributions.**   In this work, we identify a new "intrinsic LR" parameter $\lambda_e = \lambda\eta$ based on Observation 1.1. The main contributions are the following:

1. We theoretically analyse how intrinsic LR controls the evolution of effective speed of learning and how it leads to the equilibrium. This is done through incorporating BN and weight decay into the classical framework of Langevin Dynamics (Section 5).
2. Based on our theory, we empirically observed that small learning rates can perform equally well, which challenges the popular belief that good generalization requires large initial LR (Section 6).
3. Finally, we make a conjecture, called *Fast Equilibrium Conjecture*, based on mathematical intuition (Section 5) and experimental evidence (Section 6): the number of steps for reaching equilibrium in Observation 1.1 scales inversely to the intrinsic LR, as opposed to the mixing time upper bound $e^{O(1/\eta)}$ for Langevin dynamics (Bovier, 2004; Shi et al., 2020). This gives a new perspective in understanding why BN is effective in deep learning.

## 2   Related Works

**Effect of Learning Rate / Batch Size.**   The generalization issue of large batch size / small learning rate has been observed as early as (Bengio, 2012; LeCun et al., 2012). (Keskar et al., 2017) argued that the cause is that large-batch training tends to converge to sharp minima, but (Dinh et al., 2017) noted that sharp minima can also generalize well due to invariance in ReLU networks. (Li et al., 2019) theoretically analysed the effect of large learning rate in a synthetic dataset to argue that the magnitude of learning rate changes the learning order of patterns in non-homogeneous dataset. To close the generalization gap between large-batch and small-batch training, several works proposed to use a large learning rate in the large-batch training to keep the scale of gradient noise (Hoffer et al., 2017; Smith and Le, 2018; Chaudhari and Soatto, 2018; Smith et al., 2018). Shallue et al. (2019) demonstrated through a systematic empirical study that there is no simple rule for finding the optimal learning rate and batch size as the generalization error could largely depend on other training metaparameters. None of these works have found that training without a large initial learning rate can generalize equally well in presence of BN.

**Batch Normalization.** Batch Normalization is proposed in (Ioffe and Szegedy, 2015). While the original motivation is to reduce Internal Covariate Shift (ICS), (Santurkar et al., 2018) challenged this view and argued that the effectiveness of BN comes from a smoothening effect on the training objective. (Bjorck et al., 2018) empirically observed that the higher learning rate enabled by BN is responsible for the better generalization. (Kohler et al., 2019) studied the direction-length decoupling effect in BN and designed a new optimization algorithm with faster convergence for learning 1-layer or 2-layer neural nets. Another line of works focus on the effect of scale-invariance induced by BN as well as other normalization schemes. (Hoffer et al., 2018a) observed that the effective learning rate of the parameter direction is $\frac{\eta}{\|\boldsymbol{w}_t\|^2}$. (Arora et al., 2019b) identified an auto-tuning behavior for the effective learning rate and (Cai et al., 2019) gave a more quantitative analysis for linear models. In presence of BN and weight decay, (van Laarhoven, 2017) showed that the gradient noise causes the norm to grow and the weight decay causes to shrink, and the effective learning rate eventually reaches a constant value if the noise scale stays constant. (Zhang et al., 2019) validated this phenomenon in the experiments. (Li and Arora, 2020) rigorously proved that weight decay is equivalent to an exponential learning rate schedule.

## 3 Preliminaries

**Stochastic Gradient Descent and Weight Decay.** Let $\{(\boldsymbol{x}_i, y_i)\}_{i=1}^n$ be a dataset consisting of input-label pairs. In (mini-batch) stochastic gradient descent, the following is the update, where $\mathcal{B}_t \subseteq \{1, \ldots, n\}$ is a mini-batch of random samples, $\boldsymbol{w}_t$ is the vector of trainable parameters of a neural network, $\mathcal{L}(\boldsymbol{w}; \mathcal{B}) = \frac{1}{|\mathcal{B}|} \sum_{b \in \mathcal{B}} \ell_{\mathcal{B}}(\boldsymbol{w}; \boldsymbol{x}_b, y_b)$ is the average mini-batch loss (we use subscript $\mathcal{B}$ because $\ell$ can depend on $\mathcal{B}$ if BN is used) and $\eta_t$ is the learning rate (LR) at step $t$:

$$\boldsymbol{w}_{t+1} \leftarrow \boldsymbol{w}_t - \eta_t \nabla \mathcal{L}(\boldsymbol{w}_t; \mathcal{B}_t). \tag{1}$$

*Weight decay (WD)* with parameter $\lambda$ (a.k.a., adding an $\ell_2$ regularizer term $\frac{\lambda}{2} \|\boldsymbol{w}\|_2^2$) is standard in networks with BN, yielding the update:

$$\boldsymbol{w}_{t+1} \leftarrow (1 - \eta_t \lambda)\boldsymbol{w}_t - \eta_t \nabla \mathcal{L}(\boldsymbol{w}_t; \mathcal{B}_t). \tag{2}$$

**Normalization Schemes and Scale-invariance.** Batch normalization (BN) (Ioffe and Szegedy, 2015) makes the training loss invariant to re-scaling of layer weights, as it normalizes the output for every neuron (see Appendix A for details; scale-invariance emerges if the output layer is fixed). We name this property as *scale-invariance*. More formally, we say a function $f : \mathbb{R}^d \to \mathbb{R}$ is *scale-invariant* if $f(\boldsymbol{w}) = f(\alpha \boldsymbol{w}), \forall \boldsymbol{w} \in \mathbb{R}^d, \alpha > 0$. Note that scale-invariance is a general property that also holds for loss in presence of other normalization schemes (Wu and He, 2018; Salimans and Kingma, 2016; Ba et al., 2016).

Scale-invariance implies the gradient and Hessian are inversely proportional to $\|\boldsymbol{w}\|, \|\boldsymbol{w}\|^2$ respectively, meaning that the smoothness is unbounded near $\boldsymbol{w} = 0$. This can be seen by taking gradients with respect to $\boldsymbol{w}$ on both sides of $f(\boldsymbol{w}) = f(\alpha \boldsymbol{w})$:

**Lemma 3.1.** *For a scale-invariant function $f : \mathbb{R}^d \to \mathbb{R}$, $\nabla f(\alpha \boldsymbol{w}) = \frac{1}{\alpha} \nabla f(\boldsymbol{w})$ and $\nabla^2 f(\alpha \boldsymbol{w}) = \frac{1}{\alpha^2} \nabla^2 f(\boldsymbol{w})$ hold for all $\boldsymbol{w}$ and $\alpha > 0$.*

The gradient can also be proved to be perpendicular to $\boldsymbol{w}$, that is, $\langle \nabla f(\boldsymbol{w}), \boldsymbol{w} \rangle = 0$ holds for all $\boldsymbol{w}$. This property can also be seen as a corollary of Euler's Homogeneous Function Theorem. In the deep learning literature, (Arora et al., 2019b) used this in the analysis of the auto-tuning behavior of normalization schemes.

**Approximating SGD by SDE.** Define the expected loss $\mathcal{L}(\boldsymbol{w}) := \mathbb{E}_{\mathcal{B}}[\mathcal{L}(\boldsymbol{w}; \mathcal{B})]$ and the error term $\boldsymbol{\xi} := \nabla \mathcal{L}(\boldsymbol{w}; \mathcal{B}_t) - \nabla \mathcal{L}(\boldsymbol{w})$. Then we can rewrite the formula of SGD with constant LR $\eta$ as $\boldsymbol{w}_{t+1} \leftarrow \boldsymbol{w}_t - \eta(\nabla \mathcal{L}(\boldsymbol{w}_t) + \boldsymbol{\xi}_t)$. The mean of gradient noise is always 0. The covariance matrix of the gradient noise at $\boldsymbol{w}$ equals to $\boldsymbol{\Sigma}_{\boldsymbol{w}} := \mathbb{E}_{\mathcal{B}}[(\nabla \mathcal{L}(\boldsymbol{w}; \mathcal{B}) - \nabla \mathcal{L}(\boldsymbol{w}))(\nabla \mathcal{L}(\boldsymbol{w}; \mathcal{B}) - \nabla \mathcal{L}(\boldsymbol{w}))^\top]$. To approximate SGD by SDE, the classic approach is to model the gradient noise by Gaussian noise $\boldsymbol{\xi}_t \sim \mathcal{N}(\boldsymbol{0}, \boldsymbol{\Sigma}_{\boldsymbol{w}_t})$, and then take the continuous time limit to obtain the surrogate SDE for infinitesimal LR (Li et al., 2017; Cheng et al., 2019). As is done in previous works (Smith and Le, 2018; Smith et al., 2018; Chaudhari and Soatto, 2018; Shi et al., 2020), we also use this surrogate dynamics to approximate SGD with LR of any size:

$$\mathrm{d}\boldsymbol{W}_t = -\eta \left( \nabla \mathcal{L}(\boldsymbol{W}_t)\mathrm{d}t + (\boldsymbol{\Sigma}_{\boldsymbol{W}_t})^{\frac{1}{2}}\mathrm{d}\boldsymbol{B}_t \right). \tag{3}$$

Here $\boldsymbol{B}_t \in \mathbb{R}^d$ is the Wiener Process (Brownian motion), which satisfies $\boldsymbol{B}_t - \boldsymbol{B}_s \sim N(\boldsymbol{0}, (t-s)\boldsymbol{I}_d)$ conditioned on $\boldsymbol{B}_s$. When $\boldsymbol{\Sigma}_{\boldsymbol{w}}$ is $\boldsymbol{0}$ (the full-batch GD case), (3) is known as *gradient flow*.

**Folklore view of landscape exploration.** There is evidence that the training loss has many global minima (or near-minima), whose test loss values can differ radically. The basins around these global minima are separated by "hills" and only large noise can let SGD jump from one to another, while small noise will only make the network oscillate in the same basin around a minimum. The regularization effect of large LR/large noise happens because (1) sharp minima have worse generalization (2) noise prevents getting into narrow basins and thus biases exploration towards flatter basins. (But this view is known to be simplistic, as noted in many papers.)

## 4 Apparent Incompatibility between BN and Traditional View Points

In this section, we discuss how BN leads to issues with the traditional optimization view of gradient flow and SDE. This motivates our new view in Section 5. Figures 5 and 6 are deferred into Appendix D due to page limit.

**Full batch gradient descent $\neq$ gradient flow.** It's well known that if LR is smaller than the inverse of the smoothness, then trajectory of gradient descent will be close to that of gradient flow. But for normalized networks, the loss function is scale-invariant and thus provably non-smooth (i.e., smoothness becomes unbounded) around origin (Li et al., 2019). (By contrast, without WD, the SGD moves away from origin (Arora et al., 2019b) since norm increases monotonically.) We will show that this nonsmoothness is very real and makes training unstable and even chaotic for full batch SGD with any nonzero learning rate. And yet convergence of gradient flow is unaffected.

Consider a toy scale-invariant loss, $L(x, y) = \frac{x^2}{x^2+y^2}$. Since loss only depends on $x/y$, WD has no effect on it. Even with WD turned on, Gradient Flow (i.e., infinitesimal updates) will lead to monotone decrease in $|x_t/y_t|$. But Figure 5a in the appendix shows that dynamics for GD with WD are chaotic: as similar trajectories approach the origin, tiny differences are amplified and they diverge.

Modern deep nets with BN + WD (the standard setup) also exhibit instability close to zero loss. See Figures 5b and 5c, where deep nets being trained on small datasets exhibit oscillation between zero loss and high loss. In any significant period with low loss (i.e., almost full accuracy), gradient is small but WD continues to reduce the norm, and resulting non-smoothness leads to large increase in loss.

**Problems with random walk/SDE view of SGD.** The standard story about the role of noise in deep learning is that it turns a deterministic process into a geometric random walk in the landscape, which can in principle explore the landscape more thoroughly, for instance by occasionally making loss-increasing steps. Rigorous analysis of this walk is difficult since the mathematics of real-life training losses is not understood. But assuming the noise in SDE is a fixed Gaussian, the stationary distribution of the random walk can be shown to be the familiar Gibbs distribution over the landscape. See (Shi et al., 2020) for a recent account, where SDE is shown to converge to equilibrium distribution in $e^{O(C/\eta)}$ time for some term $C$ depending upon loss function. This convergence is extremely slow for small LR $\eta$ and thus way beyond normal training budget.

Recent experiments have also suggested the walk does not reach this initialization-independent equilibrium within normal training time. Stochastic Weight Averaging (SWA) (Izmailov et al., 2018) shows that the loss landscape is nearly convex along the trajectory of SGD with a fixed hyper-parameter choice, e.g., if the two network parameters from different epochs are averaged, the test loss is lower. This reduction can go on for 10 times more than the normal training budget as shown in Figure 6. However, the accuracy improvement is a very local phenomenon since it doesn't happen for SWA between solutions obtained from different initialization, as shown in (Draxler et al., 2018; Garipov et al., 2018). This suggests the networks found by SGD within normal training budget highly depends on the initialization, and thus SGD doesn't mix in the parameter space.

Another popular view (e.g., (Izmailov et al., 2018)) believes that instead of mixing to the unique global equilibrium, the trajectory of SGD could be well approximated by a multivariate Ornstein-Uhlenbeck (OU) process around a local minimizer $\boldsymbol{W}_*$, assuming the loss surface is locally strongly convex. As the corresponding stationary point is a Gaussian distribution $\mathcal{N}(\boldsymbol{W}_*, \boldsymbol{\Sigma})$, this explains why SWA helps to reduce the training loss. However, this view is challenged by the fact that the $\ell_2$ distance between weights from epochs $T$ and $T+\Delta$ monotonically increases with $\Delta$ for every $T$ (See Figure 6b), while $\mathbb{E}[\|\boldsymbol{W}_T - \boldsymbol{W}_{T+\Delta}\|_2^2]$ should converge to the constant $2\operatorname{Tr}(\boldsymbol{\Sigma})$ as $T, \Delta \to +\infty$ in the OU process. This suggests that all these weights are correlated and haven't mixed to Gaussian.

For the case where WD is turned off, (Arora et al., 2019b) proves that the norm of weight is monotone increasing, thus the mixing in parameter space provably doesn't exist for SGD with BN.

# 5 SDE-based framework for modeling SGD on Normalized Networks

For SGD with learning rate $\eta$ and weight decay $\lambda$, we define $\lambda_e := \eta\lambda$ to be the *effective weight decay*. This is actually the original definition of weight decay (Hanson and Pratt, 1989) and is also proposed (based upon experiments) in (Loshchilov and Hutter, 2019) as a way to improve generalization for Adam and SGD. In Section 5.1, we will suggest calling $\lambda_e$ the *intrinsic learning rate* because it controls trajectory in a manner similar to learning rate. Now we can rewrite update rule (2) and its corresponding SDE as

$$\boldsymbol{w}_{t+1} \leftarrow (1 - \lambda_e)\boldsymbol{w}_t - \eta\left(\nabla\mathcal{L}(\boldsymbol{w}_t) + \boldsymbol{\xi}_t\right). \tag{4}$$

$$\mathrm{d}\boldsymbol{W}_t = -\eta\left(\nabla\mathcal{L}(\boldsymbol{W}_t)\mathrm{d}t + (\boldsymbol{\Sigma}_{\boldsymbol{W}_t})^{\frac{1}{2}}\mathrm{d}\boldsymbol{B}_t\right) - \lambda_e\boldsymbol{W}_t\mathrm{d}t. \tag{5}$$

## 5.1 SDE with Weight Decay and Normalization

When the loss function is scale-invariant, the gradient noise $\boldsymbol{\Sigma}_{\boldsymbol{W}}$ is inherently anisotropic and position-dependent: Lemma B.1 in the appendix shows the noise lies in the subspace perpendicular to $w$ and blows up close to the origin. To get an SDE description closer to the canonical format, we reparametrize parameters to unit norm. Define $\overline{\boldsymbol{W}}_t = \frac{\boldsymbol{W}_t}{\|\boldsymbol{W}_t\|}$, $G_t = \|\boldsymbol{W}_t\|^2$, where $\|\boldsymbol{w}\|$ stands for the $L^2$-norm of a vector $\boldsymbol{w}$. The following Lemma is proved in the appendix using Itô's Lemma:

**Theorem 5.1.** *The evolution of the system can be described as:*

$$\mathrm{d}\overline{\boldsymbol{W}}_t = -\frac{\eta}{G_t}\left(\nabla\mathcal{L}(\overline{\boldsymbol{W}}_t)\mathrm{d}t + (\boldsymbol{\Sigma}_{\overline{\boldsymbol{W}}_t})^{\frac{1}{2}}\mathrm{d}\boldsymbol{B}\right) - \frac{\eta^2}{2G_t^2}\operatorname{Tr}(\boldsymbol{\Sigma}_{\overline{\boldsymbol{W}}_t})\overline{\boldsymbol{W}}_t\mathrm{d}t \tag{6}$$

$$\frac{\mathrm{d}G_t}{\mathrm{d}t} = -2\lambda_e G_t + \frac{\eta^2}{G_t}\operatorname{Tr}(\boldsymbol{\Sigma}_{\overline{\boldsymbol{W}}_t}). \tag{7}$$

The SDE enables clean mathematical demonstration of many properties of normalization schemes. For example, dividing both sides of (7) by $\eta$ gives

$$\frac{\mathrm{d}(G_t/\eta)}{\mathrm{d}t} = -2\lambda_e \cdot \frac{G_t}{\eta} + \frac{\eta}{G_t}\operatorname{Tr}(\boldsymbol{\Sigma}_{\overline{\boldsymbol{W}}_t}). \tag{8}$$

This shows that the dynamics only depends on the ratio $G_t/\eta$, which also suggests that *initial LR is of limited importance, indistinguishable from scale of initialization.* Now define $\gamma_t := (G_t/\eta)^2$. ($\eta/G_t = \gamma_t^{-0.5}$ was called the *effective learning rate* in (Hoffer et al., 2018a; Zhang et al., 2019; Arora et al., 2019b).) This simplifies the equations:

$$\mathrm{d}\overline{\boldsymbol{W}}_t = -\gamma_t^{-1/2}\left(\nabla L(\overline{\boldsymbol{W}}_t)\mathrm{d}t + (\boldsymbol{\Sigma}_{\overline{\boldsymbol{W}}_t})^{\frac{1}{2}}\mathrm{d}\boldsymbol{B}_t\right) - \frac{1}{2\gamma_t}\operatorname{Tr}(\boldsymbol{\Sigma}_{\overline{\boldsymbol{W}}_t})\overline{\boldsymbol{W}}_t\mathrm{d}t. \tag{9}$$

$$\frac{\mathrm{d}\gamma_t}{\mathrm{d}t} = -4\lambda_e\gamma_t + 2\operatorname{Tr}(\boldsymbol{\Sigma}_{\overline{\boldsymbol{W}}_t}). \tag{10}$$

(10) can be alternatively written as the following, which shows that squared effective LR $\gamma_t$ is a running average of the norm squared of gradient noise.

$$\gamma_t = e^{-4\lambda_e t}\gamma_0 + 2\int_0^t e^{-4\lambda_e(t-\tau)}\operatorname{Tr}(\boldsymbol{\Sigma}_{\overline{\boldsymbol{W}}_\tau})d\tau. \tag{11}$$

Experimentally[2] we find that the trace of noise is approximately constant. This is the assumption of the next lemma (much weaker than assumption of fixed gaussian noise in past works).

**Lemma 5.2.** *If $\sigma^2 \leq \operatorname{Tr}(\Sigma_{\overline{\boldsymbol{W}}}) \leq (1+\epsilon)\sigma^2$ for all $\overline{\boldsymbol{W}}$ encountered in the trajectory, then*

$$\gamma_t = e^{-4\lambda_e t}\gamma_0 + (1 + O(\epsilon))\frac{\sigma^2}{2\lambda_e}\left(1 - e^{-4\lambda_e t}\right). \tag{12}$$

The lemma again suggests that the initial effective LR decided together by LR $\eta$ and norm $\|\boldsymbol{W}_0\|$ only has a temporary effect on the dynamics: no matter how large is the initial effective LR, after $O(1/\lambda_e)$ time, the effective LR $\gamma_t^{-1/2}$ always converges to the stationary value $(1 + O(\epsilon))\frac{\sigma}{\sqrt{2\lambda_e}} \propto \lambda_e^{-1/2}$.

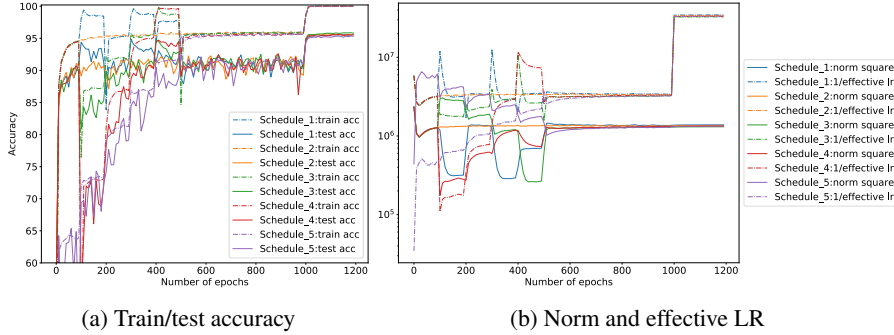

| (a) Train/test accuracy | (b) Norm and effective LR |

Figure 1: PreResNet32 trained by SGD with 5 random LR/WD schedules in the first 500 epochs converge to the same equilibrium when LR and WD factor are set the same at epoch 500 – These different trajectories exhibit similar test/train accuracy, norm and effective LR. Moreover, they achieve the same best test accuracy ($\sim 95\%$, the same as that with momentum) after decaying LR and removing WD at epoch 1000, suggesting that the equilibrium is independent of initialization. See details of the schedules in Table 1. (Appendix)

## 5.2 A conjecture about mixing time in function space

As mentioned earlier, there is evidence that SGD may take a very long time to mix in the parameter space. However, we observed that test/train errors converge in expectation soon after the norm $\|\boldsymbol{W}_t\|$ converges in expectation, which only takes $O(1/\lambda_e)$ time by our theoretical analysis. More specifically, we find experimentally that the number of steps of SGD (or the length of time for SDE) after the norm converges doesn't significantly affect the expectation of any known statistics related to the training procedure, including train/test errors, and even the output distribution on every single datapoint. This suggests the neural net reaches an "equilibrium state" in function space.

The above findings motivate us to define the notion of "equilibrium state" rigorously and make a conjecture formally. For learning rate schedule $\eta(t)$ and effective weight decay schedule $\lambda_e(t)$, we define $\nu(\mu; \lambda_e, \eta, t)$ to be the marginal distribution of $\boldsymbol{W}_t$ in the following SDE when $\boldsymbol{W}_0 \sim \mu$:

$$\mathrm{d}\boldsymbol{W}_t = -\eta(t)\left(\nabla\mathcal{L}(\boldsymbol{W}_t)\mathrm{d}t + (\boldsymbol{\Sigma}_{\boldsymbol{W}_t})^{\frac{1}{2}}\mathrm{d}\boldsymbol{B}_t\right) - \lambda_e(t)\boldsymbol{W}_t\mathrm{d}t. \tag{13}$$

For a random variable $X$, we define $P_X$ to be the probability distribution of $X$. The total variation $d_{\mathrm{TV}}(P_1, P_2)$ between two probability measures $P_1, P_2$ is defined by the supremum of $|P_1(A) - P_2(A)|$ over all measurable set $A$. Given input $\boldsymbol{x}$ and neural net parameter $\boldsymbol{w}$, we use $F(\boldsymbol{w}; \boldsymbol{x})$ to denote the class prediction of the neural net on input $\boldsymbol{x}$.

**Conjecture 5.3** (Fast Equilibrium Conjecture). Under the dynamics of (9) and (10), modern neural nets converge to the equilibrium distribution in $O(1/\lambda_e)$ time in the following sense. Given two initial distributions $\mu, \mu'$ for $\boldsymbol{W}_0$, constant learning rate and effective weight decay schedules $\lambda_e^*, \eta^*$, there exists a mixing time $T = O(1/\lambda_e^*)$ [3] such that for any input data $\boldsymbol{x}$ from some input domain $\mathcal{X}$, $d_{\mathrm{TV}}\left(P_{F(\boldsymbol{W}_t; \boldsymbol{x})}, P_{F(\boldsymbol{W}'_t; \boldsymbol{x})}\right) \approx 0$ for all $t \geq T$, where $\boldsymbol{W}_t \sim \nu(\mu; \lambda_e^*, \eta^*, t)$, $\boldsymbol{W}'_t \sim \nu(\mu'; \lambda_e^*, \eta^*, t)$.

It is worth to note that this conjecture obviously does not hold for some pathological initial distributions, e.g., all the neurons are initially dead. But we can verify that our conjecture holds for many initial distributions that can occur in training neural nets, e.g., random initialization, or the distribution after training with certain schedule for certain number of epochs. It remains a future work to theoretically identify the specific condition for this conjecture.

Interesting, we empirically find that the above conjecture still holds even if we are allowed to fine-tune the model before producing the output. This can be modeled by starting another SDE from $t \geq T$:

**Conjecture 5.4** (Fast Equilibrium Conjecture, Strong Form). Let $\tilde{\eta}(\tau), \tilde{\lambda}_e(\tau)$ be a pair of learning rate and effective weight decay schedules. Under the same conditions of Conjecture 5.3, there exists a mixing time $T = O(1/\lambda_e^*)$ such that for any input data $\boldsymbol{x}$ from some input domain $\mathcal{X}$, $d_{\mathrm{TV}}\left(P_{F(\boldsymbol{W}_{t,\tau}; \boldsymbol{x})}, P_{F(\boldsymbol{W}'_{t,\tau}; \boldsymbol{x})}\right) \approx 0$ for all $t \geq T$, where $\boldsymbol{W}_{t,\tau} \sim \nu\left(\nu(\mu; \lambda_e^*, \eta^*, t); \tilde{\lambda}_e, \tilde{\eta}, t\right)$, $\boldsymbol{W}'_{t,\tau} \sim \nu\left(\nu(\mu'; \lambda_e^*, \eta^*, t); \tilde{\lambda}_e, \tilde{\eta}, t\right)$.

In the Appendix C we provide the discrete version of the above conjecture by viewing each step of SGD as one step transition of a Markov Chain. By this means we can also extend the conjecture to SGD with momentum and even Adam.

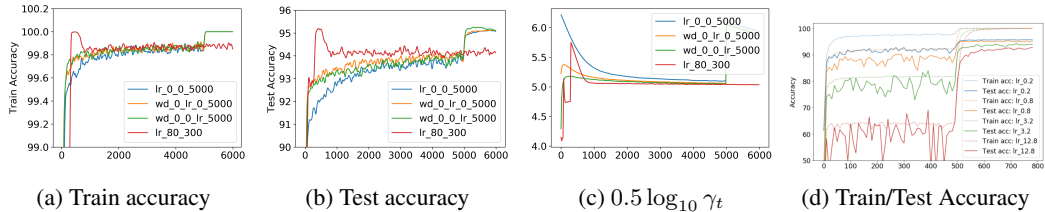

|  (a) Train accuracy | (b) Test accuracy | (c) $0.5 \log_{10} \gamma_t$ | (d) Train/Test Accuracy |

Figure 2: **(a)-(c)** Achieving SOTA test accuracy by 0.9-momentum SGD with small learning rates (the blue line). (Similar phenomenon observed for vanilla SGD, see Figure 10 in Appendix F) The initial learning rate is 0.1, initial WD factor is 0.0005. The label `wd_x_y_lr_z_u` means dividing WD factor by 10 at epoch $x$ and $y$, and dividing LR by 10 at epoch $z$ and $u$. For example, the blue line means dividing LR by 10 twice at epoch 0, i.e. using an initial LR of 0.001 and dividing LR by 10 at epoch 5000. The red line is baseline. **(d)** Equilibrium of smaller intrinsic LR leads to higher test accuracy on CIFAR after LR decay by 10. PreResNet32 trained with SGD without momentum and with WD factor 0.0005.

## 5.3 What happens in real life training – An interpretation

Let's first recap Step Decay – there are $K$ phases and LR in phase $i$ is $\bar{\eta}_i$. Below we will explain or give better interpretation for some phenomena in real life training related to Step Decay.

**Sudden increase of test error and training loss after every LR decay:** Usually here LR is dropped by something like a factor 10. As shown above, the instant effect is to reduce effective LR by a factor of 10, but it gradually equilibriates to the value $\lambda_e^{-1/2}$, which is only reduced by a factor of $\sqrt{10}$. Hence there is a slow rise in error after every drop, as observed in previous works (Zagoruyko and Komodakis, 2016; Zhang et al., 2019; Li and Arora, 2020). This rise could be beneficial since it coincides with equilibrium state in function space.

**Intrisic LR and the final LR decay step:** However, the final LR decay needs to be treated differently. It is customary do early stopping, that finish very soon after the final LR decay, when accuracy is best. The above paragraph can help to explain this by decomposing the training after the final LR decay into two stage. In the first stage, the effective LR is very small, so the dynamics is closer to the classical gradient flow approximation, which can settle into a local basin. In the second stage, the effective LR increases to the stationary value and brings larger noise and worse performance. This decomposition also applies to earlier LR decay operations, but the phenomenon is more significant for the final LR decay because the convergence time $O(1/\lambda_e)$ is much longer.

Since each phase in Step decay except the last one is allowed to reach equilibrium, the above conjecture suggests the generalization error of Step Decay schedule only depends on the intrinsic LR for its last equilibrium, namely the second-to-last phase. Thus, Step Decay could be abstracted into the following general *two-phase training* paradigm, where the only hyper-parameter of SGD that affects generalization is the intrinsic LR, $\lambda_e$:

1. **SDE Phase.** Reach the equilibrium of (momentum) SGD with $\lambda_e$ (as fast as possible).

2. **Gradient Flow Phase.** Decay the learning rate by a large constant, e.g., 10, and set $\lambda_e = 0$. Train until loss is zero.

The above training paradigm says for good generalization, what we need is only *reaching the equilibrium of small (intrinsic) LR and then decay LR and stop quickly*. In other words, the initial large LR should not be necessary to achieve high test accuracy. Indeed our experiments show that networks trained directly with the small intrinsic LR, though necessarily for much longer due to slower mixing, also achieve the same performance. See Figure 2 for SGD with momentum and Figure 10 for vanilla SGD.

**So what's the benefit of early large learning rates?** Empirically we observed that initial large (intrinsic) LR leads to a faster convergence of the training process to the equilibrium. See the red lines in Figure 2. A natural guess for the reason is that directly reaching the equilibrium of small intrinsic LR from the initial distribution is slower than to first reaching the equilibrium of a larger intrinsic LR and then the equilibrium of the target small intrinsic LR. This has been made rigorous for the mixing time of SDE in parameter space (Shi et al., 2020). In our setting, we show in Appendix E that this argument makes sense at least for norm convergence: the initial large LR reduces the gap between the current norm and the stationary value corresponding to the small LR, in a much shorter time. In Figure 8, we show that the early large learning rate is crucial for the learnability of normalized networks with initial distributions with extreme magnitude. Intriguingly, though without a theoretical analysis, early large learning rate experimentally (see Figure 10c) accelerates norm convergence and convergence to equilibrium even with momentum.

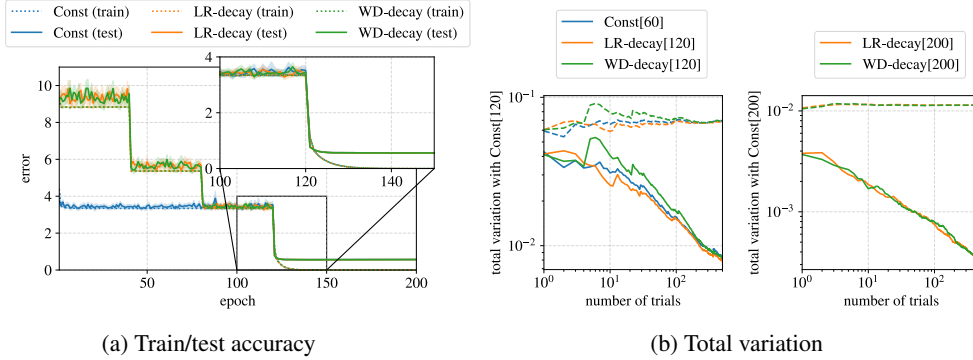

|           |                 |
|-----------|-----------------|
| (a) Train/test accuracy | (b) Total variation |

Figure 3: A simple 4-layer CNN trained on MNIST with three schedules converge to the same equilibrium after intrinsic LRs become equal at epoch 81. **(a)** The train/test errors (averaged over 500 trials) are almostly the same from epoch 81. **(b)** We estimate the total variation between the empirical distribution of the predictions on test images for neural nets trained with schedule `Const` and other schedules for 120/200 epochs (solid lines). The estimated value decreases with the number of trials. For comparison, the dashed lines are the sum of averaged test errors of each pair of training processes, which can be seen as baselines since the sum is the total variation when the set of images that lead to wrong predictions for the two training processes are completely different.

**But is it worth waiting for the equilibrium of small (intrinsic) LR?** In Figure 2d we show that different equilibrium does lead to different performance after the final LR decay. Given this experimental result we speculate the basins of different scales in the optimization landscape seems to be nested, i.e., a larger basin can contain multiple smaller basins of different performances. And reaching the equilibrium of a smaller intrinsic LR seems to be a stronger regularization method, though it also costs much more time.

**Batch size and linear scaling rule:** Recall the batch loss is $\mathcal{L}(\boldsymbol{w}; \mathcal{B}) = \frac{1}{|\mathcal{B}|} \sum_{b \in \mathcal{B}} \ell_{\mathcal{B}}(\boldsymbol{w}; \boldsymbol{x}_b, y_b)$. If $\ell_{\mathcal{B}}$ is independent of $\mathcal{B}$, such as GroupNorm or LayerNorm is used instead of BN, we have $\Sigma_{\boldsymbol{w}}^B = \frac{1}{B}\Sigma_{\boldsymbol{w}}^1$, where $\Sigma_{\boldsymbol{w}}^B$ is the noise covariance when the batch size is $B$. Therefore, let $\boldsymbol{W}_t^{B,\eta}$ denote the solution in Equation (3), we have $\boldsymbol{W}_t^{B,\eta} = \boldsymbol{W}_{Bt}^{1,\frac{\eta}{B}}$, given that the initialization are the same, i.e. $\boldsymbol{W}_0^{B,\eta} = \boldsymbol{W}_0^{1,\frac{\eta}{B}}$. In other words, up to a time rescaling, doubling the batch size $B$ is equivalent to halving down LR $\eta$ for all losses in the SDE regime, a.k.a. *linear scaling rule* (Goyal et al., 2017), in which case it can be shown that $\frac{\lambda_e}{B}$ alone determines the equilibrium of SDE. However, this analysis is less general, e.g., it doesn't work for BN, especially when batch size goes to 0, as $\Sigma_{\boldsymbol{w}}^B$ can be significantly different from $\frac{1}{B}\Sigma_{\boldsymbol{w}}^1$ due to the noise in batch statistics. Thus we treat batch size $B$ as a fixed hyper-parameter in this paper.

# 6 Experimental Evidence of Theory

## 6.1 Equilibrium is independent of the initial distribution

In this subsection we aim to show that the equilibrium only depends on the intrinsic LR, $\lambda_e = \eta\lambda$, and is independent of the initial distribution of the weights and individual values of $\eta$ and $\lambda$.

**MNIST Experiments.** We use a simple 4-layer CNN for MNIST. To highlight the effect of scale-invariance, we make the CNN scale-invariant by fixing the last linear layer as well as the affine parameters in every BN. Figure 3a shows the train/test errors for three different schedules, `Const`, `LR-decay` and `WD-decay`. Each error curve is averaged over 500 independent runs, where we call each run as a *trial*. `Const` initiates the training with $\eta = 0.1$ and $\lambda = 0.1$. `LR-decay` initiates the training with 4 times larger LR and decreases LR by a factor of 2 every 40 epochs. `WD-decay` initiates the training with 4 times larger WD and decreases WD by a factor of 2 every 40 epochs. All these three schedules share the same intrinsic LR from epoch 81 to 120, and thus reach the same train/test errors in this phase as we have conjectured. Moreover, after we setting $\eta = 0.01$ and $\lambda = 0$ at epoch 121 for fine-tuning, all the schedules show the same curve of decreasing train/test errors, which verifies the strong form of our conjecture.

Figure 3b measures the total variation between predictions of neural nets trained for 120 and 200 epochs with different schedules. Given a pair of distributions $\mathcal{W}, \mathcal{W}'$ of neural net parameters (e.g., the distributions of neural net parameters after training with `LR-decay` and `WD-decay` for 200 epochs), we enumerate each input image $x$ from the test set and compute the total variation between the empirical distributions $\mathcal{D}_x, \mathcal{D}'_x$ of the class prediction on $x$ for weights sampled from $\mathcal{W}, \mathcal{W}'$,

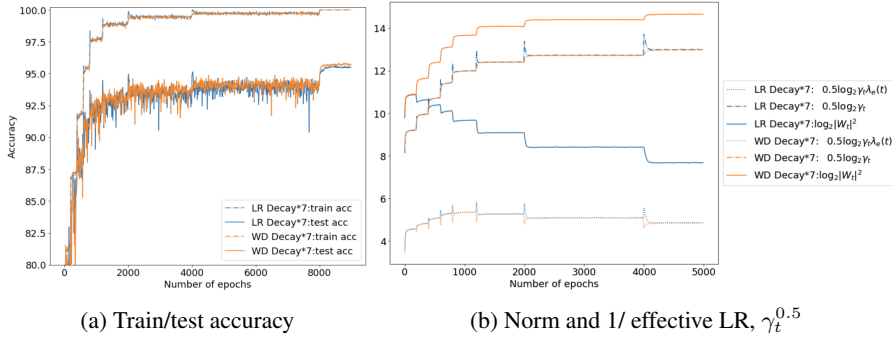

| (a) Train/test accuracy | (b) Norm and 1/ effective LR, $\gamma_t^{0.5}$ |

Figure 4: Smaller intrinsic LR takes longer time to stabilize its norm. We train two PreResNet32 by SGD with the same initial LR, $\eta = 3.2$ and WD factor, $\lambda = 0.0005$ without momentum. The LR/ WD factor are divided by 2 at epoch 200, 400, 600, 800, 1200, 2000 and 4000 respectively. Still, the networks share almost the same effective LR and train/test accuracy for most of the time. The best test accuracies for both are achieved by removing WD and dividing LR by 10 at epoch 8000.

where $\mathcal{D}_x, \mathcal{D}'_x$ are estimated via Monte Carlo. Figure 3b shows that the average total variation over test inputs decrease with the number of trials, again suggestting mixing happens in the function space.

**CIFAR-10 Experiments.** We use PreResNet32 for CIFAR10 with data augmentation and the batch size is 128. We modify the downsampling part according to the Appendix C in (Li et al., 2019) and fix the last layer and $\gamma, \beta$ in every BN, to ensure the scale invariance. In Figure 1 we focus on the comparison between the performance of the networks within and after leaving the equilibrium, where the networks are initialized differently via different LR/WD schedules before switching to the same intrinsic LR. We repeat this experiment with VGG16 on CIFAR-10 (Figure 12 in Appendix F) and PreResNet32 on CIFAR-100 (Figure 13 in Appendix F) in appendix and get the same results. A direct comparison between the effect of LR and WD can be found in Figure 4.

## 6.2 Reaching Equilibrium only takes $O(1/(\lambda\eta))$ steps

In Figure 4 we show that convergence of norm is a good measurement for reaching equilibrium, and it takes longer time for smaller intrinsic LR $\lambda_e$. The two networks are trained with the same sequence of intrinsic learning rates, where the first schedule (blue) decays LR by 2 at epoch , and the second schedule decays WD factor by 2 at the same epoch list. Note that the effective LR almost has the same trend as the training accuracy. Since in each phase, the effective LR $\gamma_t^{-0.5} \propto \|\boldsymbol{W}_t\|^{-2}$, we conclude that the convergence of norm suggests SGD reaches the equilibrium.

In Figure 11 (Appendix F) we provide experimental evidence that the mixing time to equilibrium in function space scales to $\frac{1}{\eta\lambda}$. Note in Equation (12), the convergence of norm also depends on the initial value. Thus in order to reduce the effect of initialization on the time towards equilibrium, we use the setting of Figure 3 in (Li et al., 2019), where we first let the networks with the same architecture reach the equilibrium of different intrinsic LRs, and we decay the LR by 10 and multiplying the WD factor by 10 simultaneously. In this way the intrinsic LR is not changed and the equilibrium is still the same. However, the effective LR is perturbed far away from the equilibrium, i.e. multiplied by 0.1. And we measure how long does it takes SGD to recover the network back to the equilibrium and we find it to be almost linear in $1/\lambda\eta$.

## 7 Conclusion and Open Questions

We pointed that use of normalization in today's state-of-art architectures today leads to a mismatch with traditional mathematical views of optimization. To bridge this gap we develop the mathematics of SGD + BN + WD in scale-invariant nets, in the process identifying a new hyper-parameter "intrinsic learning rate", $\lambda_e = \eta\lambda$, for which appears to determine trajectory evolution and network performance after reaching equilibrium. Experiments suggest time to equilibrium in *function space* is only $O(\frac{1}{\lambda_e})$, dramatically lower than the usual exponential upper bound for mixing time in parameter space. Our *fast equilibrium conjecture* about this may guide future theory. The conjecture suggests a more general two-phase training paradigm, which could be potentially interesting to practitioners and lead to better training.

Our theory shows that convergence of norm is a good sign for having reached equilibrium. However, we still lack a satisfying measure of the progress of training, since empirical risk is not good. Finally, it would be good to *understand why reaching equilibrium helps regularization*.

## Acknowledgement

This work is supported from NSF, ONR, Simons Foundation, Schmidt Foundation, Mozilla Research, Amazon Research, DARPA, SRC and Microsoft Research.

## Broader Impact

The observation of this paper may help understanding the generalization of deep learning and make hyper-parameter tuning easier for both researchers and practitioners.

## Footnotes

[2]Figures 4 and 11 shows that after a certain length of time the relationship $\gamma_t^{1/2} \propto \lambda_e^{-1/2}$ holds approximately, up to a small multiplicative constant. Since $\gamma_t$ is the running average of $\operatorname{Tr}(\Sigma_{\overline{\boldsymbol{W}}})$, the magnitude of the noise, it suggests for different regions of the landscape explored by SGD with different intrinsic LR $\lambda_e$, the noise scales don't differ a lot.

[3] Here we assume the both initial weight norm and the initial LR are of constant magnitude. Otherwise there will be a multiplicative $\log \gamma_0$ factor in the mixing time, as indicated by Equation (11). This $\log$ dependency can usually be ignored in practice, unless the initial weight norm or LR are extremely large or small. See Figure 8.

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
