[Supplementary Material]

# Reconciling Modern Deep Learning with Traditional Optimization Analyses: The Intrinsic Learning Rate

**Zhiyuan Li**[*]
Princeton University
zhiyuanli@cs.princeton.edu

**Kaifeng Lyu** [*]
Tsinghua University
vfleaking@gmail.com

**Sanjeev Arora**
Princeton University & IAS
arora@cs.princeton.edu

## Abstract

Recent works (e.g., (Li and Arora, 2020)) suggest that the use of popular normalization schemes (including Batch Normalization) in today's deep learning can move it far from a traditional optimization viewpoint, e.g., use of exponentially increasing learning rates. The current paper highlights other ways in which behavior of normalized nets departs from traditional viewpoints, and then initiates a formal framework for studying their mathematics via suitable adaptation of the conventional framework namely, modeling SGD-induced training trajectory via a suitable stochastic differential equation (SDE) with a noise term that captures gradient noise. This yields: (a) A new "intrinsic learning rate" parameter that is the product of the normal learning rate $\eta$ and weight decay factor $\lambda$. Analysis of the SDE shows how the effective speed of learning varies and equilibrates over time under the control of intrinsic LR. (b) A challenge—via theory and experiments—to popular belief that good generalization requires large learning rates at the start of training. (c) New experiments, backed by mathematical intuition, suggesting the number of steps to equilibrium (in function space) scales as the inverse of the intrinsic learning rate, as opposed to the exponential time convergence bound implied by SDE analysis. We name it the *Fast Equilibrium Conjecture* and suggest it holds the key to why Batch Normalization is effective.

## 1 Introduction

The training of modern neural networks involves Stochastic Gradient Descent (SGD) with an appropriate learning rate schedule. The formula of SGD with weight decay can be written as:
$$\boldsymbol{w}_{t+1} \leftarrow (1 - \eta_t \lambda)\boldsymbol{w}_t - \eta_t \nabla \mathcal{L}_t(\boldsymbol{w}_t),$$
where $\lambda$ is the weight decay factor (or $L^2$-regularization coefficient), $\eta_t$ and $\nabla \mathcal{L}_t(\boldsymbol{w}_t)$ are the learning rate and batch gradient at the $t$-th iteration.

Traditional analysis shows that SGD approaches a stationary point of the training loss if the learning rate is set to be sufficiently small depending on the smoothness constant and noise scale. In this viewpoint, if we reduce the learning rate by a factor 10, the end result is the same, and just takes 10 times as many steps. SGD with very tiny step sizes can be thought of as Gradient Descent (GD) (i.e., gradient descent with full gradient), which in the limit of infinitesimal step size approaches Gradient Flow (GF).

However, it is well-known that using only small learning rates or large batch sizes (while fixing other hype-parameters) may lead to worse generalization (Bengio, 2012; Keskar et al., 2017). From this one concludes that finite (not too small) learning rate —alternatively, noise in the gradient estimate, or small batch sizes— play an important role in generalization, and many authors have suggested that the noise helps avoid sharp minima (Hochreiter and Schmidhuber, 1997; Keskar et al., 2017; Li et al.,

---

[*]These authors contribute equally.

2018; Izmailov et al., 2018; He et al., 2019). Formal understanding of the effect involves modeling SGD via a Stochastic Differential Equation (SDE) in the continuous time limit (Li and Tai, 2019):

$$d\boldsymbol{W}_t = -\eta(t)\lambda\boldsymbol{W}_t dt - \eta(t)\nabla\mathcal{L}(\boldsymbol{W}_t)dt + \eta(t)\boldsymbol{\Sigma}_{\boldsymbol{W}_t}^{1/2}d\boldsymbol{B}_t,$$

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

[4]Strictly speaking, NTK initialization is a re-parametrization of Kaiming initialization, rather than a different initialization method, as the additional multiplier indeed changes the architecture.

[5]As discussed earlier, NTK initialization has larger weight norm. For 0.001 kaiming initialization, the reason is more subtle: the initial norm are indeed super small, thus leading to huge initial gradient, and therefore the norm grows quickly in the first few iterations.

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

# A   Batch Normalization

Batch normalization (BN) (Ioffe and Szegedy, 2015) is one of the most commonly-used normalization schemes. Given a mini-batch of inputs $\{z_b\}_{b\in\mathcal{B}}$ from the last layer, a batch normalization layer first normalizes the inputs by subtracting the mean $\mu_{\mathcal{B}} := \frac{1}{|\mathcal{B}|} \sum_{b\in\mathcal{B}} z_b$ and dividing the variance $\sigma_{\mathcal{B}}^2 = \frac{1}{|\mathcal{B}|} \sum_{b\in\mathcal{B}} (z_b - \mu_{\mathcal{B}})^2$, and then applies a linear transformation with trainable parameters $\gamma$ and $\beta$:

$$\mathrm{BN}(z_b) = \gamma \frac{z_b - \mu_{\mathcal{B}}}{\sigma_{\mathcal{B}}} + \beta.$$

Typically BN is placed between the linear transformation and activation function. This makes the loss invariant to the re-scaling of weights in the linear transformation preceding the BN. If we fix the weights in the last linear layer as suggested by (Hoffer et al., 2018b) and put BN after every linear transformation, then the loss is invariant to all its parameters (see Appendix C of (Li and Arora, 2020) for more details).

# B   Missing derivation and proofs

For scale-invariant loss function $\mathcal{L}(\boldsymbol{w}; \mathcal{B})$, we have the following lemma on the covariance matrix of gradient noise:

**Lemma B.1.** *If $\mathcal{L}(\boldsymbol{w}; \mathcal{B})$ is scale-invariant with respect to $\boldsymbol{w}$, then*

1. $\boldsymbol{\Sigma}_{c\boldsymbol{w}} = c^{-2}\boldsymbol{\Sigma}_{\boldsymbol{w}}$ *for any $c > 0$.*

2. $\|\boldsymbol{\Sigma}_{\boldsymbol{w}}^{1/2}\boldsymbol{w}\|_2 = \boldsymbol{w}^{\top}\boldsymbol{\Sigma}_{\boldsymbol{w}}\boldsymbol{w} = 0.$

*Proof.* Note that the expectation of scale-invariant functions is scale-invariant, so $\mathcal{L}(\boldsymbol{w})$ is scale-invariant. The first bullet can be proved by combining the definition of $\boldsymbol{\Sigma}_{c\boldsymbol{w}}$ and

$$\nabla\mathcal{L}(c\boldsymbol{w}; \mathcal{B}) - \nabla\mathcal{L}(c\boldsymbol{w}) = \frac{1}{c}\left(\nabla\mathcal{L}(\boldsymbol{w}; \mathcal{B}) - \nabla\mathcal{L}(\boldsymbol{w})\right).$$

For the second bullet, by homogeneity we have $\langle\nabla\mathcal{L}(\boldsymbol{w}; \mathcal{B}), \boldsymbol{w}\rangle = 0$ and $\langle\nabla\mathcal{L}(\boldsymbol{w}), \boldsymbol{w}\rangle = 0$, so $\langle\nabla\mathcal{L}(\boldsymbol{w}; \mathcal{B}) - \nabla\mathcal{L}(\boldsymbol{w}), \boldsymbol{w}\rangle = 0$. This implies $\boldsymbol{w}^{\top}\boldsymbol{\Sigma}_{\boldsymbol{w}}\boldsymbol{w} = \mathbb{E}[\langle\nabla\mathcal{L}(\boldsymbol{w}; \mathcal{B}) - \nabla\mathcal{L}(\boldsymbol{w}), \boldsymbol{w}\rangle^2] = 0.$  □

To prove Theorem 5.1, we will need to use the Itô's lemma, which is stated below:

**Lemma B.2** (Itô's Lemma). *Suppose $\boldsymbol{X}_t = (X_t^1, X_t^2, \ldots, X_t^d)$ is a vector of Itô's processes s.t.*

$$\mathrm{d}\boldsymbol{X}_t = \boldsymbol{\mu}_t \mathrm{d}t + \boldsymbol{G}_t \mathrm{d}\boldsymbol{B}_t,$$

*we have for any twice differentiable function $f$,*

$$df(t, \mathbf{X}_t) = \frac{\partial f}{\partial t}\, dt + (\nabla_{\mathbf{X}} f)^T\, d\mathbf{X}_t + \frac{1}{2}\,(d\mathbf{X}_t)^T\,(H_{\mathbf{X}} f)\, d\mathbf{X}_t,$$

$$= \left\{\frac{\partial f}{\partial t} + (\nabla_{\mathbf{X}} f)^T \boldsymbol{\mu}_t + \frac{1}{2}\,\mathrm{Tr}\left[\mathbf{G}_t^T\left(\nabla_{\mathbf{X}}^2 f\right)\mathbf{G}_t\right]\right\} dt + (\nabla_{\mathbf{X}} f)^T \mathbf{G}_t\, d\mathbf{B}_t$$

Recall the following original SDE in the space of $\boldsymbol{W}$. Below we will prove Theorem 5.1 by Itô's Lemma.

$$\mathrm{d}\boldsymbol{W}_t = -\eta\left(\nabla\mathcal{L}(\boldsymbol{W}_t)\mathrm{d}t + (\boldsymbol{\Sigma}_{\boldsymbol{W}_t})^{\frac{1}{2}}\mathrm{d}\boldsymbol{B}_t\right) - \lambda_e\boldsymbol{W}_t\mathrm{d}t.$$

**Theorem 5.1.** *The evolution of the system can be described as:*

$$\mathrm{d}\overline{\boldsymbol{W}}_t = -\frac{\eta}{G_t}\left(\nabla\mathcal{L}(\overline{\boldsymbol{W}}_t)\mathrm{d}t + (\boldsymbol{\Sigma}_{\overline{\boldsymbol{W}}_t})^{\frac{1}{2}}\mathrm{d}\boldsymbol{B}_t\right) - \frac{\eta^2}{2G_t^2}\,\mathrm{Tr}(\boldsymbol{\Sigma}_{\overline{\boldsymbol{W}}_t})\overline{\boldsymbol{W}}_t\mathrm{d}t \tag{6}$$

$$\frac{\mathrm{d}G_t}{\mathrm{d}t} = -2\lambda_e G_t + \frac{\eta^2}{G_t}\,\mathrm{Tr}(\boldsymbol{\Sigma}_{\overline{\boldsymbol{W}}_t}). \tag{7}$$

*Proof for Theorem 5.1.* We can prove (6) and (7) by Itô's Lemma. For (7), note that $G_t = \|\boldsymbol{W}_t\|^2$, we have

$$
\begin{aligned}
\mathrm{d}G_t &= 2\boldsymbol{W}_t^\top \mathrm{d}\boldsymbol{W}_t + \mathrm{d}\boldsymbol{W}_t^\top \mathrm{d}\boldsymbol{W}_t \\
&= 2\left\langle \boldsymbol{W}_t, -\eta\left(\nabla\mathcal{L}(\boldsymbol{W}_t)\mathrm{d}t + (\boldsymbol{\Sigma}_{\boldsymbol{W}_t})^{\frac{1}{2}}\mathrm{d}\boldsymbol{B}_t\right) - \lambda_e \boldsymbol{W}_t \mathrm{d}t\right\rangle + \eta^2 \operatorname{Tr}(\boldsymbol{\Sigma}_{\boldsymbol{W}_t})\mathrm{d}t.
\end{aligned}
$$

By scale-invariance and Lemma B.1, $\langle \boldsymbol{W}_t, \nabla\mathcal{L}(\boldsymbol{W}_t)\rangle = 0$, $\left\langle \boldsymbol{W}_t, (\boldsymbol{\Sigma}_{\boldsymbol{W}_t})^{\frac{1}{2}}\mathrm{d}\boldsymbol{B}_t\right\rangle = 0$, $\operatorname{Tr}(\boldsymbol{\Sigma}_{\boldsymbol{W}_t}) = \frac{1}{G_t}\operatorname{Tr}(\boldsymbol{\Sigma}_{\overline{\boldsymbol{W}}_t})$. So we can simplify the formula to conclude that

$$
\mathrm{d}G_t = -2\lambda_e G_t \mathrm{d}t + \frac{\eta^2}{G_t}\operatorname{Tr}(\boldsymbol{\Sigma}_{\overline{\boldsymbol{W}}_t})\mathrm{d}t.
$$

For (6), let $\boldsymbol{v} \in \mathbb{R}^d$ be an arbitrary vector, then

$$
\begin{aligned}
\mathrm{d}\left\langle \boldsymbol{v}, \overline{\boldsymbol{W}}_t\right\rangle &= \left\langle \frac{\partial \left\langle \boldsymbol{v}, \overline{\boldsymbol{W}}_t\right\rangle}{\partial \boldsymbol{W}_t}, \mathrm{d}\boldsymbol{W}_t\right\rangle + \frac{1}{2}(\mathrm{d}\boldsymbol{W}_t)^\top \frac{\partial^2 \left\langle \boldsymbol{v}, \overline{\boldsymbol{W}}_t\right\rangle}{(\partial \boldsymbol{W}_t)^2}\mathrm{d}\boldsymbol{W}_t \\
&= \left\langle \frac{1}{\|\boldsymbol{W}_t\|}\left(\boldsymbol{v} - \left\langle \boldsymbol{v}, \overline{\boldsymbol{W}}_t\right\rangle\overline{\boldsymbol{W}}_t\right), \mathrm{d}\boldsymbol{W}_t\right\rangle + \frac{\eta^2}{2}\operatorname{Tr}\left((\boldsymbol{\Sigma}_{\boldsymbol{W}_t})^{\frac{1}{2}}\frac{\partial^2 \left\langle \boldsymbol{v}, \overline{\boldsymbol{W}}_t\right\rangle}{(\partial \boldsymbol{W}_t)^2}(\boldsymbol{\Sigma}_{\boldsymbol{W}_t})^{\frac{1}{2}}\right)\mathrm{d}t
\end{aligned}
$$

By scale-invariance and Lemma B.1, $\langle \boldsymbol{W}_t, \nabla\mathcal{L}(\boldsymbol{W}_t)\rangle = 0$, $(\boldsymbol{\Sigma}_{\boldsymbol{W}_t})^{\frac{1}{2}}\boldsymbol{W}_t = \boldsymbol{0}$, which means the column span of $\boldsymbol{\Sigma}_{\boldsymbol{W}_t}$ is orthogonal to $\boldsymbol{W}_t$. Thus we can apply Lemma B.3 below and get

$$
\operatorname{Tr}\left((\boldsymbol{\Sigma}_{\boldsymbol{W}_t})^{\frac{1}{2}}\frac{\partial^2 \left\langle \boldsymbol{v}, \overline{\boldsymbol{W}}_t\right\rangle}{(\partial \boldsymbol{W}_t)^2}(\boldsymbol{\Sigma}_{\boldsymbol{W}_t})^{\frac{1}{2}}\right) = -\frac{\boldsymbol{v}^\top \overline{\boldsymbol{W}}_t}{\|\boldsymbol{W}_t\|^2}\operatorname{Tr}\left(\boldsymbol{\Sigma}_{\boldsymbol{W}_t}\right).
$$

Then we have

$$
\mathrm{d}\left\langle \boldsymbol{v}, \overline{\boldsymbol{W}}_t\right\rangle = \left\langle \frac{\boldsymbol{v}}{\|\boldsymbol{W}_t\|}, -\eta\left(\nabla\mathcal{L}(\boldsymbol{W}_t)\mathrm{d}t + (\boldsymbol{\Sigma}_{\boldsymbol{W}_t})^{\frac{1}{2}}\mathrm{d}\boldsymbol{B}_t\right)\right\rangle - \frac{\eta^2}{2\|\boldsymbol{W}_t\|^2}\left\langle \overline{\boldsymbol{W}}_t, \boldsymbol{v}\right\rangle\operatorname{Tr}(\boldsymbol{\Sigma}_{\boldsymbol{W}_t})\mathrm{d}t.
$$

By scale-invariance, this can be simplified to the following formula:

$$
\mathrm{d}\left\langle \boldsymbol{v}, \overline{\boldsymbol{W}}_t\right\rangle = -\frac{\eta}{\|\boldsymbol{W}_t\|^2}\left\langle \boldsymbol{v}, \nabla\mathcal{L}(\overline{\boldsymbol{W}}_t)\mathrm{d}t + (\boldsymbol{\Sigma}_{\overline{\boldsymbol{W}}_t})^{\frac{1}{2}}\mathrm{d}\boldsymbol{B}_t\right\rangle - \frac{\eta^2}{2\|\boldsymbol{W}_t\|^4}\operatorname{Tr}(\boldsymbol{\Sigma}_{\overline{\boldsymbol{W}}_t})\left\langle \boldsymbol{v}, \overline{\boldsymbol{W}}_t\right\rangle\mathrm{d}t,
$$

which proves (6), since $\boldsymbol{v}$ is arbitrary. $\qquad\square$

**Lemma B.3.** *For any twice differentiable function $g : \mathbb{R}^d \to \mathbb{R}$, the Hessian matrix of $f(\boldsymbol{w}) := g(\frac{\boldsymbol{w}}{\|\boldsymbol{w}\|})$ satisfies that, $\forall \boldsymbol{v}^\top \boldsymbol{w} = 0$,*

$$
\boldsymbol{v}^\top \nabla^2 f(\boldsymbol{w})\boldsymbol{v} = \frac{1}{\|\boldsymbol{w}\|^2}\left(\boldsymbol{v}^\top \nabla^2 g(\bar{\boldsymbol{w}})\boldsymbol{v} - \bar{\boldsymbol{w}}^\top \nabla g(\bar{\boldsymbol{w}})\|\boldsymbol{v}\|^2\right). \tag{14}
$$

*where $\bar{\boldsymbol{w}} = \frac{\boldsymbol{w}}{\|\boldsymbol{w}\|}$.*

*Proof.* For any $\boldsymbol{v} \in \mathbb{R}^d$ s.t. $\boldsymbol{v}^\top \boldsymbol{w} = 0$, we define $h(\lambda) = f(\boldsymbol{w} + \lambda\|\boldsymbol{w}\|\boldsymbol{v})$. Then by definition, $\|\boldsymbol{w}\|^2 \boldsymbol{v}^\top \nabla^2 f(\boldsymbol{w})\boldsymbol{v} = h''(0)$.

On the other hand, note $\boldsymbol{v}^\top \boldsymbol{w} = 0$, we have

$$
\begin{aligned}
\frac{\boldsymbol{w} + \lambda\|\boldsymbol{w}\|\boldsymbol{v}}{\|\boldsymbol{w} + \lambda\|\boldsymbol{w}\|\boldsymbol{v}\|} &= \frac{\boldsymbol{w} + \lambda\|\boldsymbol{w}\|\boldsymbol{v}}{\|\boldsymbol{w}\|}\frac{1}{\sqrt{1 + \lambda^2\|\boldsymbol{v}\|^2}} \\
&= (\bar{\boldsymbol{w}} + \lambda\boldsymbol{v})(1 - \frac{\lambda^2\|\boldsymbol{v}\|^2}{2} + O(\lambda^4)) \\
&= \bar{\boldsymbol{w}} + \lambda\boldsymbol{v} - \frac{\lambda^2\|\boldsymbol{v}\|^2}{2}\bar{\boldsymbol{w}} + O(\lambda^3).
\end{aligned}
$$

Thus

$$h(\lambda) = g(\bar{\boldsymbol{w}} + \lambda\boldsymbol{v} - \frac{\lambda^2 \|\boldsymbol{v}\|^2}{2}\bar{\boldsymbol{w}} + O(\lambda^3))$$

$$= g(\bar{\boldsymbol{w}}) + \lambda\nabla g(\bar{\boldsymbol{w}})^\top\boldsymbol{v} + \frac{\lambda^2}{2}\left(\boldsymbol{v}^\top\nabla^2 g(\bar{\boldsymbol{w}})\boldsymbol{v} - \|\boldsymbol{v}\|^2\nabla g(\bar{\boldsymbol{w}})^\top\boldsymbol{v}\right) + O(\lambda^3),$$

from which we conclude $h''(0) = \boldsymbol{v}^\top\nabla^2 g(\bar{\boldsymbol{w}})\boldsymbol{v} - \|\boldsymbol{v}\|^2\nabla g(\bar{\boldsymbol{w}})^\top\boldsymbol{v}$. □

*Proof for Lemma 5.2.* By (11), we can upper bound and lower bound $\gamma_t$ by

$$\gamma_t \le e^{-4\lambda_e t}\gamma_0 + 2\int_0^t e^{-4\lambda_e(t-\tau)}(1+\epsilon)\sigma^2 d\tau \le e^{-4\lambda_e t}\gamma_0 + \frac{1}{2\lambda_e}(1-e^{-4\lambda_e t})(1+\epsilon)\sigma^2 d\tau.$$

$$\gamma_t \ge e^{-4\lambda_e t}\gamma_0 + 2\int_0^t e^{-4\lambda_e(t-\tau)}\sigma^2 d\tau \ge e^{-4\lambda_e t}\gamma_0 + \frac{1}{2\lambda_e}(1-e^{-4\lambda_e t})\sigma^2 d\tau.$$

Therefore, we have $\gamma_t = e^{-4\lambda_e t}\gamma_0 + (1+O(\epsilon))\frac{\sigma^2}{2\lambda_e}\left(1-e^{-4\lambda_e t}\right)$. □

**Connection to Exp LR schedule:** (Li and Arora, 2020) shows that

$$\boldsymbol{w}_{t+1} \leftarrow \boldsymbol{w}_t - \eta(1-\lambda_e)^{-2t}\left(\nabla\mathcal{L}(\boldsymbol{w}_t) + \boldsymbol{\xi}_t\right)$$

yields the same trajectory in function space as Equation (4) for scale invariant loss $\mathcal{L}$. In fact, they also correspond to the same surrogate SDE Equations (9) and (10), where the exponent in the rate schedule is the intrinsic LR.

**Lemma B.4.** *The following SDE with exponential LR is equivalent to Equations* (9) *and* (10)*, where* $\gamma_t = \frac{\|\boldsymbol{W}_t\|^4 e^{-4\lambda_e t}}{\eta^2}$.

$$\mathrm{d}\boldsymbol{W}_t = -e^{2\lambda_e t}\eta\left(\nabla\mathcal{L}(\boldsymbol{W}_t)\mathrm{d}t + (\boldsymbol{\Sigma}_{\boldsymbol{W}_t})^{\frac{1}{2}}\mathrm{d}\boldsymbol{B}_t\right).$$

*Proof.* By Itô's Lemma, let $\boldsymbol{U}_t = e^{-\lambda_e t}\boldsymbol{W}_t$, we have

$$\mathrm{d}\boldsymbol{U}_t = -\lambda_e\boldsymbol{U}_t\mathrm{d}t + e^{-\lambda_e t}\mathrm{d}\boldsymbol{W}_t$$

$$= -\lambda_e\boldsymbol{U}_t\mathrm{d}t + e^{-\lambda_e t}\left(\nabla\mathcal{L}(\boldsymbol{W}_t)\mathrm{d}t + (\boldsymbol{\Sigma}_{\boldsymbol{W}_t})^{\frac{1}{2}}\mathrm{d}\boldsymbol{B}_t\right)$$

$$= -\lambda_e\boldsymbol{U}_t\mathrm{d}t + \nabla\mathcal{L}(\boldsymbol{U}_t)\mathrm{d}t + (\boldsymbol{\Sigma}_{\boldsymbol{U}_t})^{\frac{1}{2}}\mathrm{d}\boldsymbol{B}_t,$$

where the last step is by scale-invariance.

Note $\boldsymbol{U}_t$ has the same direction as $\boldsymbol{W}_t$, i.e. $\overline{\boldsymbol{U}_t} = \overline{\boldsymbol{W}}_t$, we can apply Theorem 5.1 to get Equations (6) and (7), and thus get Equations (9) and (10), with $\gamma_t = \frac{\|\boldsymbol{U}_t\|^4}{\eta^2} = \frac{\|\boldsymbol{W}_t\|^4 e^{-4\lambda_e t}}{\eta^2}$. □

# C   Extension to Other Optimization Algorithms

## C.1   Momentum

In this subsection we use momentum SGD as an example to show how does the discrete version of the fast equilibrium conjecture look like. Throughout this section we will assume all the momentum factors are constant, and we only care about the role of LR $\eta$ and WD factor $\lambda$ in the discrete dynamics.

For fixed LR $\eta$ and WD $\lambda$, the formula of SGD with momentum can be written as follows:

$$\boldsymbol{v}_{t+1} \leftarrow \beta\boldsymbol{v}_t + (\nabla\mathcal{L}(\boldsymbol{w}_t; \mathcal{B}_t) + \lambda\boldsymbol{w}_t)$$

$$\boldsymbol{w}_{t+1} \leftarrow \boldsymbol{w}_t - \eta\boldsymbol{v}_{t+1},$$

which is also equivalent to

$$\boldsymbol{w}_{t+1} - \boldsymbol{w}_t = \beta(\boldsymbol{w}_t - \boldsymbol{w}_{t-1}) - \eta(\nabla\mathcal{L}(\boldsymbol{w}_t; \mathcal{B}_t) + \lambda\boldsymbol{w}_t).$$

We can decouple the effect of WD from SGD by replacing $\eta\lambda$ by $\lambda_e$:

$$\boldsymbol{w}_{t+1} - (1 + \beta - \lambda_e)\boldsymbol{w}_t + \beta\boldsymbol{w}_{t-1} = -\eta\nabla\mathcal{L}(\boldsymbol{w}_t; \mathcal{B}_t).$$

By scale invariance of $\mathcal{L}$, letting $\boldsymbol{w}_t' = \frac{\boldsymbol{w}_t}{\sqrt{\eta}}$, we have

$$\boldsymbol{w}_{t+1}' - (1 + \beta - \lambda_e)\boldsymbol{w}_t' + \beta\boldsymbol{w}_{t-1}' = -\nabla\mathcal{L}(\boldsymbol{w}_t'; \mathcal{B}_t),$$

which means the effect of $\eta$ in the new parametrization is no more than rescaling the initialization. This motivates as to define $\lambda_e = \eta\lambda$ as the effective WD, or intrinsic LR.

Unlike vanilla SGD, the evolution of norm for momentum SGD is more complicated. However, a folklore intuition is that, if the gradient of loss $\mathcal{L}$ changes slowly, one can approximate momentum SGD by vanilla SGD with LR $\frac{\eta\lambda}{1-\gamma}$. Therefore, we propose the following discrete version of fast equilibrium conjecture.

For LR schedule $\eta(t)$ and WD schedule $\lambda(t)$, we define $\nu(\mu; \lambda, \eta, t)$ to be the marginal distribution of $(\boldsymbol{w}_t, \boldsymbol{v}_t)$ in the following dynamical system when $(\boldsymbol{w}_0, \boldsymbol{v}_0) \sim \mu$:

$$\boldsymbol{v}_{t+1} \leftarrow \beta\boldsymbol{v}_t + (\nabla\mathcal{L}(\boldsymbol{w}_t; \mathcal{B}_t) + \lambda(t)\boldsymbol{w}_t)$$
$$\boldsymbol{w}_{t+1} \leftarrow \boldsymbol{w}_t - \eta(t)\boldsymbol{v}_{t+1}$$

**Conjecture C.1** (Fast Equilibrium Conjecture for Momentum). For SGD with momentum, modern neural nets converge to the equilibrium distribution in $O(1/\lambda_e)$ time in the following sense. Given two initial distributions $\mu, \mu'$ for $\boldsymbol{w}_0$, constant LR and effective WD schedules $\lambda^*, \eta^*$, there exists a mixing time $T = O(1/\lambda_e^*)$, where $\lambda_e^* = \eta^*\lambda^*$, such that for any input data $\boldsymbol{x}$ from some input domain $\mathcal{X}$,

$$d_{\mathrm{TV}}\left(P_{F(\boldsymbol{w}_t; \boldsymbol{x})}, P_{F(\boldsymbol{w}_t'; \boldsymbol{x})}\right) \approx 0,$$

for all $t \geq T$, where $(\boldsymbol{w}_t, \boldsymbol{v}_t) \sim \nu(\mu; \lambda^*, \eta^*, t)$, $(\boldsymbol{w}_t', \boldsymbol{v}_t') \sim \nu(\mu'; \lambda^*, \eta^*, t)$.

Moreover, let $\tilde{\eta}(\tau), \tilde{\lambda}(\tau)$ be a pair of LR and WD schedules, then there exists a mixing time $T = O(1/\lambda_e^*)$ such that for any input data $\boldsymbol{x}$ from some input domain $\mathcal{X}$,

$$d_{\mathrm{TV}}\left(P_{F(\boldsymbol{w}_{t,\tau}; \boldsymbol{x})}, P_{F(\boldsymbol{w}_{t,\tau}'; \boldsymbol{x})}\right) \approx 0$$

for all $t \geq T$, where $\boldsymbol{w}_{t,\tau} \sim \nu\left(\nu(\mu; \lambda^*, \eta^*, t); \tilde{\lambda}, \tilde{\eta}, t\right)$, $\boldsymbol{w}_{t,\tau}' \sim \nu\left(\nu(\mu'; \lambda^*, \eta^*, t); \tilde{\lambda}, \tilde{\eta}, t\right)$.

## C.2 Adam

---

**Algorithm 1** <span style="background-color:#e8a5c8">Adam with L$_2$ regularization</span> and <span style="background-color:#c3d86e">Adam with decoupled weight decay (AdamW)</span> [Copied from (Loshchilov and Hutter, 2019) ]

---

1: **given** $\alpha = 0.001, \beta_1 = 0.9, \beta_2 = 0.999, \epsilon = 10^{-8}, \lambda \in \mathbb{R}$
2: **initialize** time step $t \leftarrow 0$, parameter vector $\boldsymbol{\theta}_{t=0} \in \mathbb{R}^n$, first moment vector $\boldsymbol{m}_{t=0} \leftarrow \mathbf{0}$, second moment vector $\boldsymbol{v}_{t=0} \leftarrow \mathbf{0}$, schedule multiplier $\eta_{t=0} \in \mathbb{R}$
3: **repeat**
4:     $t \leftarrow t + 1$
5:     $\nabla f_t(\boldsymbol{\theta}_{t-1}) \leftarrow$ SelectBatch$(\boldsymbol{\theta}_{t-1})$ {select batch and return the corresponding gradient}
6:     $\boldsymbol{g}_t \leftarrow \nabla f_t(\boldsymbol{\theta}_{t-1})$ <span style="background-color:#e8a5c8">$+\lambda\boldsymbol{\theta}_{t-1}$</span>
7:     $\boldsymbol{m}_t \leftarrow \beta_1\boldsymbol{m}_{t-1} + (1 - \beta_1)\boldsymbol{g}_t$ {here and below all operations are element-wise}
8:     $\boldsymbol{v}_t \leftarrow \beta_2\boldsymbol{v}_{t-1} + (1 - \beta_2)\boldsymbol{g}_t^2$
9:     $\hat{\boldsymbol{m}}_t \leftarrow \boldsymbol{m}_t/(1 - \beta_1^t)$ {$\beta_1$ is taken to the power of $t$}
10:    $\hat{\boldsymbol{v}}_t \leftarrow \boldsymbol{v}_t/(1 - \beta_2^t)$ {$\beta_2$ is taken to the power of $t$}
11:    $\eta_t \leftarrow$ SetScheduleMultiplier$(t)$ {can be fixed, decay, or also be used for warm restarts}
12:    $\boldsymbol{\theta}_t \leftarrow \boldsymbol{\theta}_{t-1} - \eta_t\left(\alpha\hat{\boldsymbol{m}}_t/(\sqrt{\hat{\boldsymbol{v}}_t} + \epsilon)\ \text{<span style="background-color:#c3d86e">}+\lambda\boldsymbol{\theta}_{t-1}\text{</span>}\right)$
13: **until** *stopping criterion is met*
14: **return** optimized parameters $\boldsymbol{\theta}_t$

---

**Connection to AdamW:** (Loshchilov and Hutter, 2019) found that using the parametrization of $\lambda_e = \eta\lambda$ achieves better generalization and a more separable hyper-parameter search space for SGD

and Adam, which are named SGDW and AdamW respectively. So far we have justified the role of intrinsic LR for SGD(W). The theorem below shows that the notion of the intrinsic LR also holds for AdamW, while the learning rate has no more power than initialization scale.

**Theorem C.2.** *For fixed scale-invariant losses $\{f_t(\boldsymbol{w})\}_{t=1}$, constant schedule multiplier $\eta_t \equiv \eta$ and $\epsilon = 0$, multiplying the initial weight $\boldsymbol{W}_0$ and the learning rate $\alpha$ by the same constant $C$ would not change the trajectory of AdamW (Appendix C.2) in function space.*

**Remark C.3.** *The notation of LR is slightly different in (Loshchilov and Hutter, 2019) than in the main paper, where $alpha$ is LR and $\eta_t$ is the* SCHEDULE MULTIPLIER. *By using schedule multiplier, AdamWAppendix C.2 can decay LR and WD factor simultaneously. This notation is only used for the statement of the above theorem and its proof.*

*Proof.* The proof is based on induction. It suffices to prove that for two history $\{\boldsymbol{\theta}_t\}_{t=0}$ and $\{\boldsymbol{\theta}'_t\}_{t=0}$ satisfying that $C\boldsymbol{\theta}_t = \boldsymbol{\theta}'_t$, for all $0 \le t \le T$ and some $C > 0$, and evolving with $\alpha$ and $\alpha' = C\alpha$ respectively, the following holds:

$$C\boldsymbol{\theta}_{T+1} = \boldsymbol{\theta}'_{T+1}.$$

Note that by scale invariance of $f_t$, $\boldsymbol{g}_t = C\boldsymbol{g}'_t, \forall 1 \le t \le T$, therefore by definition $\alpha \hat{\boldsymbol{m}}_T / \sqrt{\hat{\boldsymbol{v}}_T} = \alpha \hat{\boldsymbol{m}}'_T / \sqrt{\hat{\boldsymbol{v}}'_T}$, i.e. it's independent of scaling of the history. We now can conclude that

$$C\boldsymbol{\theta}_{T+1} = C\boldsymbol{\theta}_T - \eta C\alpha \hat{\boldsymbol{m}}_T / \sqrt{\hat{\boldsymbol{v}}_T} - \eta C\boldsymbol{\theta}_T = \boldsymbol{\theta}'_T - \eta\alpha' \hat{\boldsymbol{m}}'_T / \sqrt{\hat{\boldsymbol{v}}'_T} - \eta\boldsymbol{\theta}'_T = \boldsymbol{\theta}_{T+1},$$

which completes the proof. $\square$

# D  Supplementary Figures for Section 4

In this section, we provide experimental evidence, Figures 5 and 6, for incompatibilities omitted in Section 4.

(a) GD with WD on $L(x, y)$  (b) Train accuracy  (c) Norm

Figure 5: WD makes GD on scale-invariant loss unstable and chaotic, for both the toy model and PreResNet32. **(a)** The toy model is trained GD with LR 0.1, WD 0.5 and two initial points near zero loss. The initial points are very close to each other. **(b)(c)** *Convergence never truly happens* for PreResNet32 trained on sub-sampled CIFAR10 containing 1000 images with full-batch GD, WD $5 \times 10^{-4}$, LR 1.6 (without momentum). PreResNet32 can easily get 100% training accuracy but is unable to stay long. WD is turned off at epoch 30000.

# E  Discussion on the Benefit of Early Large Intrinsic LR

Fast equilibrium conjecture says that the equilibrium can be reached in $O(1/\lambda_e)$ steps for all reasonable initializations. Indeed, Equation (12) indicates that there is also a logarithmic dependency on $\frac{\gamma_t}{\gamma_0}$, i.e., if the initial effective LR is far from the effective LR at equilibrium, then the mixing time can be larger by a multiplicative constant compared to good initial effective LRs. Below we show this constant improvement coming from a good initialization matters a lot for real life training (meaning the training budget is limited), and the usage of initial large intrinsic LR helps SGD to reach a better initialization for the final phase, and thus allow faster mixing to the final equilibrium.

(a) Test accuracy

(b) Pairwise $\ell_2$ distance

Figure 6: Stochastic Weight Averaging improves the test accuracy of PreResNet32 trained vanilla SGD on CIFAR10. In particular, test accuracy is improved by $4\%$ by simply averaging a network with any other network along the same trajectory, suggesting the trajectory is still local. However, the distance between parameters keeps increasing. As a comparison, the average parameter norm (over epoch 2000-8000) is around 39, which has exactly the same magnitude as the pairwise distance, indicating the langevin diffusion view around strongly convex local optimum in (Izmailov et al., 2018) may not suffice to explain the success of SWA. Similar phenomenon is observed for monetum SGD, see Figure 7.

(a) Test accuracy

(b) Pairwise $\ell_2$ distance

Figure 7: Stochastic Weight Averaging improves the test accuracy of PreResNet32 trained with momentum SGD on CIFAR10.

**The benefit of early large intrinsic learning rates.** In this section we give experimental evidence that how the fast equilibrium conjecture led by BatchNorm + WD makes the Step Decay training schedule robust to various different initialization methods. In detail, we compare the following 4 types of initialization: Neural Tangent Kernel (NTK) initialization (Jacot et al., 2018; Arora et al., 2019a), Kaiming initialization (He et al., 2015) , Kaiming initialization multiplied by 1000 and Kaiming initialization multiplied by 0.001. In Figure 8 we show that the initial large (intrinsic) learning rate in Step Decay is very necessary to ensure SGD reach the equilibrium of small (intrinsic) LR within the normal training budget, and thus achieving good test accuracy.

**Comparison between the above four initialization.** Briefly speaking, these methods are quite similar as they all initialize each parameter by i.i.d. Gaussian, and the only difference is the variance of the gaussian distribution in each layer. For Kaiming initialization, the variance is roughly $\frac{1}{N}$, and for NTK initialization, the variance is always $O(1)$ but there is an additional multiplier of $O(\frac{1}{\sqrt{N}})$ per layer, where $N$ is the number of the input channels/neurons that layer. [4] Note that the NTK initialization and Kaiming initialization are always the same in function space. Due the scale invariance led by BatchNorm, all the scaled version of Kaiming initialization are the same as the original Kaiming initialization in function space.

(a) Train/test Accuracy

(b) Norm and effective LR

Figure 8: The large initial intrinsic LR (as well as large WD factor) helps achieve high test accuracy within normal training budget consistently for different initialization methods. The training curve and convergence time to equilibrium for large initial LR is robust even to the extreme small/large initializations. PreResNet32 trained by momentum SGD with initial LR $0.1$ on CIAFR10 with 4 different initialization methods, 2 different WD values, and 2 different LR schedules. Each LR schedule divides its LR by 10 twice at epoch [80,120] (the normal schedule) or epoch [0,120] (meaning starting with a 10 times smaller LR, $0.01$). The red line and orange line performs much better than their counterparts (without initial large LR) when not using standard Kaiming Initialization. Still, the red line even outperforms orange line a lot when the initialization are extremely large or small, due to the effect of large intrinsic LR brought by large WD factor. This justifies the argument in Section 5 that the equilibrium of small intrinsic LR is much closer to that of large intrinsic LR, than some arbitrary random initialization. This is very clear from the view of norm convergence. See (b).

**A Theoretical Analysis on Norm Convergence.** Although the convergence of norm is not equivalent to the convergence in function space, analysing the convergence of norm can provide insights into how large LR helps training. Now we theoretically analyse the effect of early large LR on the convergence rate of norm. We compare the following two processes with the same initial norm squared $G_0$:

1. Train the neural net with LR $\eta$;
2. Train the neural net with intrinsic LR $K\eta$, then decay it to $\eta$ after the norm converges.

For simplicity, we consider the case that $\frac{\sigma^2}{2\eta\lambda} = 1$, which means $\gamma_t$ in the first process eventually converges to $1 + O(\epsilon)$; other cases can be transformed to this case by re-scaling the initialization.

For the first process, $\gamma_t = G_0/\eta^2$ initially. By Lemma 5.2, $\gamma_t$ converges to $1 + O(\epsilon)$ in

$$O\left(\frac{1}{\eta\lambda}\max\left\{\ln\frac{G_0}{\eta^2}, 1\right\}\right)$$

time. For the second process, $\gamma_t = G_0/(K^2\eta^2)$ initially, and $\gamma_t$ first converges to $(1 + O(\epsilon))\frac{1}{K}$ in $O\left(\frac{1}{K\eta\lambda}\max\left\{\ln\frac{G_0}{K\eta^2}, 1\right\}\right)$ time. After LR decay, $\gamma_t$ instantly becomes $(1 + O(\epsilon))K$ as $\gamma_t$ is inversely proportional to LR squared. Then we only need another $O(\frac{1}{\eta\lambda}\ln K)$ time to make the effective LR converges again. Overall, the second process takes

$$O\left(\frac{1}{K\eta\lambda}\max\left\{\ln\frac{G_0}{K\eta^2}, 1\right\} + \frac{1}{\eta\lambda}\ln K\right) \tag{15}$$

time. Comparing the second process with the first process, we can see that the large initial LR reduces the dependence of convergence time on the initial norm. It is worth to note that $\frac{G_0}{\eta^2}$ is typically larger than $K$ (which equals to 10) in Figure 8. Therefore, a large initial LR also leads to faster convergence time without tuning the initialization scale.

**Explanation for different convergence rates in Figure 8:** The 4 settings about LR schedules and WD can be interpreted using Equation (15) as the choices of $(K, \lambda)$. Let $\eta = 0.01$, $K = 1$ means starting with $\eta = 0.01$, while $K = 10$ means starting with the default LR, 0.1, which is 10 times larger than $\eta$. For the rest 3 initializations other than kaiming initialization, from Figure 8, we can see that the initial norm are all exponentially large[5], making $\ln\frac{G_0}{K\eta^2}$ a large constant. Thus the total steps of training has to be $\Omega(\frac{1}{K\eta\lambda}\ln\frac{G_0}{K\eta^2})$ for the effective learning rate to grow and the training to proceed. This could also be seen directly from the ratio of the slopes of the log norm square, which is $1 : 10 : 5 : 50$.

## F  Supplementary Figures and Tables for Section 6

### F.1  Equilibrium is Independent of Initialization

This subsection provides supplementary materials to justify that the equilibrium is independent of initialization.

Table 1 shows the LR and WD of each random schedule in Figure 1. Figure 12 and Figure 13 are experiments in similar settings as Figure 1 to show that the equilibrium is independent of initialization for VGG16 on CIFAR-100.

We also validate our claim in the case that we initialize the training with a single possible initial point $\boldsymbol{w}_0$ in a similar setting as Figure 3. That is, we first randomly sample a parameter from the distribution for random initialization, and use it to initialize CNNs in all the independent runs for estimating the equilibrium. Figure 9 shows that CNNs still converge to the equilibrium even if the initial parameter $\boldsymbol{w}_0$ is fixed to the same random sample.

| Epoch | 0 | 100 | 200 | 300 | 400 | 500 | 1000 |
|---|---|---|---|---|---|---|---|
| Schedule_1 | - | LR/4 | LR×4 | LR/4 | LR×2 | LR×2 | LR/10,WD = 0 |
| Schedule_2 | - | - | - | - | - | - | LR/10,WD = 0 |
| Schedule_3 | - | LR×4 | LR/2 | LR/2 | LR/4 | LR×4 | LR/10,WD = 0 |
| Schedule_4 | - | LR,WD×4 | LR,WD/2 | LR,WD/2 | LR,WD/4 | LR,WD×4 | LR/10,WD = 0 |
| Schedule_5 | LR×32 | LR/2 | LR/2 | LR/2 | LR/2 | LR/2 | LR/10,WD = 0 |

Table 1: LR/WD Schedules in Figure 1. All the schedules have the same initial LR = 0.4 and classic WD = 0.0005. The batch size is 128 and momentum is turned off.

Figure 9: CNNs trained on MNIST converge to the equilibrium even if the initial parameter $w_0$ is fixed to some random sample. We estimate the total variation between the empirical distribution of the predictions on test images for neural nets trained with schedule Const with fixed $w_0$ and other schedules for 120/200 epochs (solid lines). The estimated value decreases with the number of trials. The dashed lines are the sum of averaged test errors of each pair of training processes which can be seen as baselines.

## F.2  Equilibrium Can be Reached in $O(1/\lambda\eta)$ Steps

In this subsection we provide more experimental evidence that the mixing time to equilibrium in function space scales to $\frac{1}{\eta\lambda}$. Note in (12), the convergence of norm also depends on the initial value. Thus in order to reduce the effect of initialization on the time towards equilibrium, we use the setting of Figure 3 in (Li et al., 2019), where we first let the networks with the same architecture reach the equilibrium of different intrinsic LRs, and we decay the LR by 10 and multiplying the WD factor by 10 simultaneously. In this way the intrinsic LR is not changed and the equilibrium is still the same. However, the effective LR is perturbed far away from the equilibrium, i.e. multiplied by 0.1. And we measure how long does it takes SGD to recover the network back to the equilibrium and we find it to be almost linear in $1/\lambda\eta$.

(a) Train accuracy        (b) Test accuracy        (c) Effective LR, $\gamma_t^{-1/2}$

Figure 10: Achieving SOTA test accuracy by 0.9-momentum SGD with small learning rates (the blue line). The initial learning rate is 0.1, initial WD factor is 0.0005. The label `wd_x_y_lr_z_u` means dividing WD factor by 10 at epoch $x$ and $y$, and dividing LR by 10 at epoch $z$ and $u$. For example, the blue line means dividing LR by 10 twice at epoch 0, i.e. using an initial LR of 0.01 and dividing LR by 10 at epoch 5000. The red line is baseline.

(a) Train/test accuracy

(b) Norm and effective LR

Figure 11: VGG16 was trained on CIFAR10 with BN + SGD and different intrinsic LRs. Then LR and WD were changed while maintaining their product (i.e., intrinsic LR). Number of steps to reach equilibrium again was measured. It scales inversely with intrinsic LR, supporting Fast Equilibrium Conjecture.

(a) Train/test accuracy

(b) Norm and effective LR

Figure 12: VGG16 trained by SGD on CIAFR10 with 5 random LR/WD schedules in Table 1, same as that in Figure 1. These different trajectories exhibit similar test/train accuracy, norm and effective LR after switching to the same intrinsic LR at epoch 500. Moreover, they achieve the same best test accuracy ($\sim 94\%$) after decaying LR and removing WD at epoch 1000. This again supports the conjecture that the equilibrium is independent of initialization.

(a) Train/test accuracy

(b) Norm and effective LR

Figure 13: PreResNet32 trained by SGD on CIAFR100 with 5 random LR/WD schedules in Table 1, same as that in Figure 1. These different trajectories exhibit similar test/train accuracy, norm and effective LR after switching to the same intrinsic LR at epoch 500. Moreover, they achieve the same best test accuracy ($\sim 78\%$) after decaying LR and removing WD at epoch 1000, thus supporting the conjecture that the equilibrium is independent of initialization.