[Reviews · NeurIPS 2020]

Review 1

Summary and Contributions: UPDATE: The authors adequately addressed my questions. My score remains the same. This paper studies the dynamics of a scale-invariant function optimized under gradient descent with weight decay (e.g. neural networks with batchnorm). It begins with a nice survey of recent results as well as apparent contradictions from these theories and empirical phenomena. Then it proposes a new SDE model of such optimization dynamics, from which it derives a few implications, such as the intrinsic learning rate and its relation to effective weight decay. Based on empirical evidence, the paper raises two conjectures that such dynamics converge rapidly to an equilibrium distribution. In the context of the SDE framework and these conjectures, the authors revisit empirical observations and re-interpret them. Finally, an array of experiments were done to verify the theoretical insights.

Strengths: The paper is very nicely written, and I learned a lot from reading it. The survey of prior results is coherent and adequately motivates the questions investigated in this paper. The SDE analysis, though nothing sophisticated, is clean, and this simultaneous simplicity and clarity should be applauded. One of the main claims in this paper is that, when training a normalized neural network for a long time, the end result only depends on the intrinsic learning rate, and not the initial conditions. This is well supported by the experiments in the paper, and I’m relatively convinced (though I have questions about the time needed to reach this equilibrium, below). I personally have wondered about why dropping the learning rate will suddenly increase test error then settle down again. (Though not sure whether this is a particularly important question in the grand scheme). This paper gives a very nice explanation of this, which I find satisfying. I also find interesting the interpretation of normalized SGD training as a combination of a SDE phase and a gradient flow phase.

Weaknesses: Among several, your paper makes two concrete predictions: 1. When dropping learning rate by 10, the intrinsic learning rate drops by 10 immediately (this is obvious), but it eventually converges to sqrt(10) 2. Reaching equilibrium takes O(1/\lambda_e) steps. I’d like to see experiments measuring and verifying them, or if your results are already in the paper, have them be more prominent, and linked to where these predictions are discussed. For example, I’d like to see a plot that plots 1/\lambda_e vs “step to convergence”, which should be linear if your prediction is correct. Other questions I have 1. Recent works suggest that gradient noise may be heavy tailed [1]. How would that change your theoretical insights? 2. Recent works indicate batchnorm causes severe gradient explosion in deep networks at initialization [2]. Would your experiments still hold in such deep networks? (say, depth 100 BN MLP) 3. What happens if you drop the learning rate before equilibrium (I assume this is common in practice)? Is the performance better or worse? Do the networks in practice reach equilibrium in the typical training time frame? 4. You show that the performance of small LR equilibrium is better than large LR equilibrium. If I just want to capture the best performance right after the learning rate drop, is small LR or large LR preferable? 5. Another increasing popular lr schedule is linear warmup and linear decay. What does this theory say about them? In my experience, in such a schedule, the performance is much more sensitive to learning rate than the weight decay, so some nontrivial correction seems to be needed to the theory here, at least to the prediction that “\lambda_e determines everything”. 6. What is the “number of trials” in fig 2(b)? 7. Does the theoretical prediction of “\lambda_e determines everything” hold when there’s momentum? 8. In fig 4(b), why do the solid curves (log weight norm) diverge from each other? Typos 1. Line 194: effective LR is gamma^-1/2 not gamma^1/2 2. Line 237: “equilibrium” typo (missing “i”) 3. Line 300: should it be norm^-1/2 instead of norm^-2? [1] Simsekli et al. A Tail-Index Analysis of Stochastic Gradient Noise in Deep Neural Networks. https://arxiv.org/abs/1901.06053 [2] Yang et al. A Mean Field Theory of Batch Normalization. https://arxiv.org/abs/1902.08129

Correctness: Yes

Clarity: Yes

Relation to Prior Work: Yes

Reproducibility: Yes

Additional Feedback:


Review 2

Summary and Contributions: The paper extends the SDE perspective of SGD to incorporate the combined influence of weight decay and scale invariance (which arises when normalization methods are used). This leads to a number of interesting insights, most strikingly that the equilibrium distribution of constant learning rate SGD in the long time limit will be governed by an effective learning rate = learning rate*weight decay coefficient. Empirically, the authors observe evidence that the mixing time into this equilibrium distribution is surprisingly fast.

Strengths: The authors tackle an important problem and identify some surprising and potentially practically useful conclusions. I believe this could prove to be an important contribution to the field which others may build on.

Weaknesses: In some places the authors over-claim, and a number of the authors comments felt misleading (see comments below). However I think many of these issues are easily resolved.

Correctness: I have not checked the derivation in detail.

Clarity: The paper reads well and is mostly easy to follow despite the technical content. However the figures are poorly presented and difficult to interpret.

Relation to Prior Work: The authors give a good discussion of prior work.

Reproducibility: Yes

Additional Feedback: Edit: I thank the authors for their response and am happy with their comments. 1) The presentation of the original SGD=SDE missing the key step, which is to identify that the minibatch noise is inverse in the batch size. This allows us to identify a temperature T=learning rate/B, and consequently we are able to reduce the learning rate to approach the SDE limit by simultaneously reducing the batch size. (of course, a key criticism of the SDE limit is that the batch size is bounded by 1). The role of batch size is also not clarified in the scale invariant SDE? 2) The authors argue that their results indicate that large learning rates do not generalize well, but a better presentation would be to say that they show that large effective learning rates generalize well. While one can make the naive learning rate small by changing the weight decay, this is no different to making the learning rate small by changing the batch size. It does not contradict the claim that finite learning rates aid generalization. 3) The authors claim that the fast equilibrium conjecture explains the benefits of batchNorm. This statement is too strong. Note that the scale invariant SDE also applies to layerNorm/instanceNorm, yet these methods generalize significantly worse than BN. 4) Additionally, the primary benefits of BN arise in resNets, and previous work has shown that this occurs because BN preserves signal propagation at initialization in resNets. This property is not captured by the analysis here. 5) Usually the SDE is defined by identifying the learning rate = the timestep dt. Here the authors introduce dt explicitly. Does this alter the analysis? 6) The authors suggest that their work will criticise the gaussian noise assumption, yet their SDE appears to assume Gaussian noise? 7) "Gradient descent not equal to gradient flow": this section appears to simply note that gradient descent with finite learning rates is not gradient flow and can be chaotic in some landcapes. Am i missing something? 8) I did not follow how the SWA experiment indicates that SGD has not equilibriated? it appears to simply indicate that SGD is fluctuating in a local minimum (consistent with equilibrium). Furthermore the authors main argument later is that SGD does equilibriate quickly (with BN)? 9) The authors make a striking claim that changing the initial learning rate is equivalent to changing the initialization scale. Are there any experiments to verify this? 10) Note that it is already recognized that the original step-wise schedules generalize poorly. Popular modern schedules (eg cosine decay) combine a large initial learning rate with rapid decay at late times. Intuitively, it is believed that this rapid decay is beneficial precisely because it prevents equilibration, thus preserving the generalization benefit of the initial learning rate. I therefore did not find the mnist results very surprising, since the step-wise decay schedule used allows training to equilibrate after each drop. I would also encourage the authors to extend their learning rate sweep to smaller learning rates to identify the optimum. 11) The authors are correct to note that, in the step wise schedule, the large initial learning rate primarily enables fast convergence. However their proposed schedule still comprises two stages, an initial finite (ie large) learning rate stage, followed by v small learning rates to simulate gradient flow. Could they clarify whether they are arguing that gradient flow generalizes as well as finite learning rates or not? (assuming infinite compute budgets)


Review 3

Summary and Contributions: The paper analyzes modern models with BN layers that trained using an SGD optimizerwith a regime based on LR step scheduler and weight decay (WD). The paper formulatesSGD as SDE and define the intrinsic LR which is the product of the LR and WD.They show the number of steps for reaching equilibrium in scales inversely tothe intrinsic LR thus controls the model convergence. Farther more they showthat small LR  can perform equally welland if forcing the model to reach equilibrium ( which require more steps withsmall LR) we can get even better results. 

Strengths: Thepaper has strong theoretical analysis and it jointly examines the connectionbetween LR WD and BN. The paper challenge common DNN training practice andsuggest a new method to investigate model convergence. 

Weaknesses: Althoughthe paper suggests many different observations, proofs, and interpretation I foundthe paper to be not cohesive and thus hard to follow.  For instance, although mentioned in relatedwork they do not explain how intrinsic LR affects large-batch training. Also,despite the extensive amount of experiments, I found the figures hard to understandand unintuitive. 

Correctness: I didnot find any error in the derivation and the experiment setting seems fair. Iwould like to emphasize that even one experiment on a larger dataset thatemphasizes the importance of iLR would have been beneficial. 

Clarity: Theresome typos and grammar issues but I think the maid caveat is the lack of onecohesive storyline. Although mentioned in the introduction, I am not sure how theirmethod translates to other normalization schemes (beside BN), how iLR isaffected by the large-batch size, and how it explains SWA boost. Similarly, theircode is neither clean nor documented thus it is hard to understand how to runit and reproduce\extend their results. 

Relation to Prior Work: Yes.

Reproducibility: Yes

Additional Feedback: Sadly,despite the great potential of this paper has I recommend the authors to rewrite it as currently it is very hard to follow and I believe many important observations and interpretations don’t get enough attention.  --------- After Rebuttal -------- I would like to thank the authors for their answers. The response adressed my concerns. I will reaise my score to 7.


Review 4

Summary and Contributions: This work applies stochastic differential equation analysis to the the learning process of networks using batch normalization. It concludes that the final equilibrium does rely on its initial states and only on one state parameter \lambda_e that is the product of the learning rate and weight decay. The authors also conjecture that the learning time to reach equilibrium is inversely proportional to \lambda_e instead of an exponential time bound. This paper is generally interesting as it delves into the mechanisms of learning of BN and the conclusions are also of interest to the community for better designing learning strategies and algorithms. Concerns from me are mainly on its assumption part and its small-sclae experimental verification. Instead of analyzing BN, this study actually is limited to a general normalization + independent noise framework. To experimentally verify its assumption and conclusions, the authors also should provide results on larger datasets. ----------Post rebuttal comments----------------- The rebuttal has addressed my concerns and well explains itself in its framework, therefore I would like to raise my score to 7.

Strengths: This study applies stochastic differential equation to learning dynamics of networks with BN, which is of importance on the theoretical development of learning dynamics. The derivation and writing are also neat. The conclusion that the equilibrium state only depends on the "intrinsic learning rate" is also interesting to the community on the understanding of BN.

Weaknesses: 1. This study is conducted under the assumption of Wiener process and continuity limit, on which the conclusion that the equilibrium state does not rely on its initial learning rate and initialization strongly depends on. Actually the performance of weight normalization + gradient noise, which should be a better candidate for the current analysis, does not reach as high as BN. Therefore, the mechanism of learning dynamics of BN is only partially addressed in this study. 2. As for the experimental verification, only mnist and cifar-10 examples are shown here. The authors should show its effectiveness on larger datasets such as ImageNet to gain more credit. 3. As also pointed out by the author, the mixing in parameter space does not exist for optimization without weight decay since the weight norm monotonically increases. The authors should give a short discussion on the threshold of weight decay that learning falls into the "no mixing in parameter space" zone.

Correctness: This derivation of study is generally correct to my knowledge except the assumptions adopted as I stated above.

Clarity: This paper is clearly written.

Relation to Prior Work: This study has addressed its difference with the prior work.

Reproducibility: Yes

Additional Feedback:

[Author Response · NeurIPS 2020]

We thank reviewers for their thorough reading. We will fix the typos and clarify the unclear points in the next version of our paper.

**On the Batch Size.** Reviewer #2 and #4 concern about the batch size. Batch size has been an important component of past analyses. For nets with BN the direct
relationship between noise and batch size mentioned by Reviewer #2 does not hold (Though there were attempts to make that theory fit using Ghost BN). In our theory
the noise is captured in covariance matrix (in eqn 9 and 10) which could depend on the batch size. When the nets are without BN, e.g. with LN or GN, the magnitude
and trace of the covariance scales inversely proportional to the batch size $B$. Thus for SDE, intrinsic LR $\lambda_e$ and the batch size $B$ can be further grouped into $\lambda_e/B$,
which alone controls eqn 9 and 10. However, this analysis doesn't hold for the general case where BN is allowed and thus we treat batch size as a fixed hyper-parameter
like width and depth. We will clarify the connection to batch size and point out that our theory could automatically include the effect of batch size when applicable, such
as when Layer Norm and Group Norm is used.

**On the benefits of BN.** Regarding Comment 3,4 of Reviewer #2 and Comment 1 of Reviewer #5, we want to clarify we do not claim that our theory covers all the
benefits of BN. The fast equilibrium conjecture only partially explains the benefits of BN. Besides this conjecture, there are many other benefits, e.g., BN affects the
signal propagation at initialization, BN induces noise when calculating the batch statistics, and our theory clearly does not capture these benefits.

**To Reviewer #1 @ Experimental verification for the two predictions:** 1. In Figure 7(b), the dotted red line (1/effective lr) increases by 10 times and then drops by
approximately $\sqrt{10}$ slowly in 40 epochs. If we make the second phase longer, one should expect the ratio becomes closer to $\sqrt{10}$. 2. Figure 10 gives a more clear and
direct justification for the claim that reaching equilibrium takes $O(1/\lambda_e)$ steps.

**@ Q1: What if the noise is heavy-tailed?** If the gradient has unbounded second moment, then we should see the weight norm has a sudden huge increase in practice,
though with small probability. However, this is not observed in any of our settings, so it's not clear to us whether the heavy tail assumption holds for our setting.
Moreover, the recent work (`https://arxiv.org/abs/1907.03215`) suggests that when the learning rate is sufficiently small (smaller than some practical constant),
the performance of neural nets only depends on the continuous-time covariance.

**@ Q2: Do our experiments still hold if large depth causes gradient explosion at initialization?** Our theory still holds. The gradient explosion at initialization leads
to huge weight norm, which makes effective lr tiny and the nets untrainable in Yang et al., since they don't have WD, the weight norm can only become even larger.
However, in our experiments, the huge weight norm led by gradient explosion could be fixed by WD and the curves of 1/effective LR again matches our theoretic
prediction, i.e. dropping slowly after the sudden increase. We ran the experiments but due to the space limit, we cannot include the figure in the rebuttal.

**@ Q3: What happens if we drop the LR before equilibrium?** The answer depends on whether or not this is the final LR decay. In the earlier phases, decaying too
soon (i.e. before equilibrium) will increase convergence time in next phase, but similar test error is reached. For final LR decay, if it happens too soon then the test error
is hurt. A good example is the blue curve in Figure 1,(a)-(c). If the lr decay happens within normal training budget, then the blue curve will get much lower test acc.
That's why people usually think tiny learning rate generalizes poorly.

**@ Q4: Is small LR or large LR preferable?** Small LR is preferable under the practically feasible regime. See Figure 1(d). It's not clear when the lr goes to 0 since
training time becomes too large.

**@ Q5: How about continuously-varying LR schedule?** Our theory doesn't analyse continuously-varying lr schedules and we leave this for future work. We suspect
equilibrium does not exist in such settings and thus the analysis should be more complicated.

**@ Q6: What is the "number of trials" in Figure 2(b)?** We repeat the training process 500 times in Figure 2, and we call each run as a "trial".

**@ Q7: How about Momentum?** Yes. When discussing the relation between our analysis and the practice, we switch from SGD to Momentum in the experiments
(Figure 1). We also discuss the extension of this conjecture to momentum formally in Section C.1. (BTW we don't quite claim $\lambda_e$ determines everything but that it
explains a lot.)

**@ Q8: Why do weight norms diverge from each other in Fig 4(b)?** According to our theoretical predictions, the two training processes in Figure 4 should have the
same effective learning rate ($\eta/\|\mathbf{w}_t\|^2$). Since they have different learning rates ($\eta$), their norms (the solid lines) should diverge from each other.

**To Reviewer #2 @ Comment 5: Time rescaling?** It does not make a difference. Note that the phenomenon is described jointly by eqn 9 and 10. The different scaling
suggested above would still capture the same phenomenon but would not highlight at a glance that the evolution depends only on intrinsic LR.

**@ Comment 6: We seemed to complain about gaussian noise assumptions but used it anyway.** We show mathematically the noise is not an isotropic Gaussian: in
fact it's position-dependent and perpendicular to the current weight. It wasn't meant as criticism of use of Gaussians per se in modeling.

**@ Comment 7: Explain more about GD $\neq$ GF.** Our point is that GD $\neq$ GF is unavoidable for scale-invariant networks with WD. This chaotic behavior is not limited
to the toy example. See Figure 5(b),(c) in the appendix for the experiments on small CIFAR.

**@ Comment 8: Explain more about SWA.** Reviewer is referring to standard SWA which fixes initialization and seems to suggest that at the end SGD is fluctuating
around a local minimum (loosely speaking). Our experiment around line 164-165 is trying to answer the question: "Is this equilibrium independent of initialization?"
The fact that SWA over solutions obtained from different initializations harms the test accuracy suggests that the answer is no. We will make this more clear in the future
version.

**@ Comment 9: Changing initial LR = changing initialization scale?** Lemma 2.4 in (Li & Arora, 2020) [28] formally proves that changing LR is equivalent to
changing the scale of the norm for SGD and scale invariant objective. In other words, shrinking norm will not change the network in function space, and it changes the
future networks after SGD updates in the same way as increasing LR does. Related experiments could also be found in (Li & Arora, 2020) [28], such as training neural
nets with exponential increasing LR schedule could lead to the same trajectory of parameters in direction, while the norm of parameters is exponentially larger.

**@ Comment 11: Gradient flow vs finite LR?** We think this refers to text around 252. We are not proposing a 2-step schedule there. The schedule could have many
decay steps; the only point being made there is that at the end training one needs a *few* epochs of very small learning rate at the end to reach best accuracy. The final
stage does not go to equilibrium and it is customary to turn off WD. In the SDE view, turning off WD is like zeroing the effective LR. That is the sense in which it is like
gradient flow. (Another evidence is that the norm does not change much during these few epochs; see Lemma 5.2 For relation between norm growth and effective LR.)

**To Reviewer #4 @ About the storyline.** We will revise our paper to improve the presentation of our main storyline and figures. Our storyline is cohesive: we use a
suitable SDE to reconcile the modern neural networks with BN + WD and traditional optimization analysis. This yields a new parameter intrinsic LR, a challenge to the
benefit of initial large LR, and the fast equilibrium conjecture.

**To Reviewer #5 @ Comment 3: What is the range of WD that leads to no mixing in parameter space?** As long as WD is non-zero, the norm is not diverging, and
thus the training process is likely to mix in parameter space. However, it may take exponential time. The conjecture in our paper concerns the ability of SGD to mix in
functional space in polynomial time.

[Meta-Review · NeurIPS 2020]

Thank you for submitting your work to NeurIPS. All four reviewers were enthusiastic about the paper, and I am happy to accept it. In the final revision, please address reviewers' feedback. Especially, please make sure to address the reviewers' 2 remark "authors argue that their results indicate that large learning rates do not generalize well, but a better presentation would be to say that they show that large effective learning rates generalize well.". Indeed, it is somewhat a strawman argument to say that other researchers claim that small LR never generalize. It is rather that when comparing the effect of a large v a small LR (*keep other hyperparameters fixed*), the network trained with a larger learning rate tends to generalize better. Finally, please also discuss a bit more clearly relation to https://arxiv.org/abs/1706.05350. Please note that the authors also claim (without a clear theoretical argument) that LR*weight decay steers learning dynamics. It would be also useful for the reader to discuss (as concurrent work)https://arxiv.org/abs/2006.08419.